# Uncoupling apical constriction from tissue invagination

SeYeon Chung, Sangjoon Kim, Deborah J Andrew*

Department of Cell Biology, The Johns Hopkins University School of Medicine, Baltimore, United States

**Abstract** Apical constriction is a widely utilized cell shape change linked to folding, bending and invagination of polarized epithelia. It remains unclear how apical constriction is regulated spatiotemporally during tissue invagination and how this cellular process contributes to tube formation in different developmental contexts. Using *Drosophila* salivary gland (SG) invagination as a model, we show that regulation of *folded gastrulation* expression by the Fork head transcription factor is required for apicomedial accumulation of Rho kinase and non-muscle myosin II, which coordinate apical constriction. We demonstrate that neither loss of spatially coordinated apical constriction nor its complete blockage prevent internalization and tube formation, although such manipulations affect the geometry of invagination. When apical constriction is disrupted, compressing force generated by a tissue-level myosin cable contributes to SG invagination. We demonstrate that fully elongated polarized SGs can form outside the embryo, suggesting that tube formation and elongation are intrinsic properties of the SG.

## Introduction

Organs that transport gases and nutrients, as well as those producing and secreting vital hormones and enzymes, are organized as epithelial tubes, many of which arise from already polarized epithelial sheets (*Andrew and Ewald, 2010*). To enter the third dimension, a flat sheet of polarized epithelial cells bends or invaginates using either of two distinct processes–wrapping or budding (*Lubarsky and Krasnow, 2003*). During wrapping, the entire epithelial sheet folds in until its edges meet and seal to form an elongated tube, as occurs in vertebrate neural tube formation and in *Drosophila* gastrulation (*Andrew and Ewald, 2010*; *Massarwa et al., 2014*). During budding, a subset of cells extend out of the plane of the epithelium in an orthogonal direction to form a tube; this process is observed during branching morphogenesis of many organs, including the mammalian lungs and kidney, and the primary branches of the *Drosophila* trachea (*Andrew and Ewald, 2010*; *Lubarsky and Krasnow, 2003*).

A limited number of cellular processes are involved in creating three-dimensional structures, which include regulated changes in cell shape, arrangement and position, as well as oriented cell divisions and spatially restricted programmed cell death (*Andrew and Ewald, 2010*). One cell shape change associated with such tissue remodeling is apical constriction, wherein the nuclei move to a basal position in the cell and the apical domains constrict (*Martin and Goldstein, 2014*; *Sawyer et al., 2010*). In polarized epithelial cells that maintain cell-cell adhesion, apical constriction is linked to tissue folding or invagination (*Alvarez and Navascués, 1990*; *Hardin and Keller, 1988*; *Kam et al., 1991*; *Lewis, 1947*; *Sweeton et al., 1991*; *Wallingford et al., 2013*).

Non-muscle myosin II-dependent contractility generates the force that drives this cellular process. Particularly, a pulsatile actomyosin complex in the apical medial region of the cell (hereafter referred to as apicomedial myosin) has been described in tissues that undergo apical constriction (*Blanchard et al., 2010*; *Martin et al., 2009*). Studies in early *Drosophila* embryos have identified

*For correspondence: dandrew@jhmi.edu

Competing interests: The authors declare that no competing interests exist.

**eLife digest** Many organs in the human body – like the kidneys, lungs, and salivary glands – are organized as a single layer of cells that surround a hollow tube. There are a number of ways that cells can achieve this particular arrangement. In one mechanism, a small group of cells bud out of a single cell layer to become the end of a new tube or a new branch of an existing tube. Since all the cells are still connected, the first cells bring their neighbouring cells along behind them, rearranging these cells to form the walls of a tube.

In addition to changing position, the cells must change their shape to form a tube. One crucial change in cell shape is called apical constriction, and involves the side of the cell facing the inside of the tube becoming smaller than the other sides. This creates cells with a wedge-like shape that can fit together to form the curved wall of the tube, similar to shaped bricks in an archway. Apical constriction has been widely studied and is controlled by proteins that act like motors moving along protein-based filaments; however the roles of apical constriction in tube formation have not been fully explained.

Using the developing salivary glands of the fruit fly *Drosophila melanogaster*, Chung et al. confirmed that the motor protein known as myosin II controls apical constriction during tissue invagination. Further examination showed that proteins (called Fork Head and Fog) activate and localize an enzyme (Rho kinase) to control the localized accumulation of myosin II and thereby control apical constriction. Chung et al. then showed that salivary glands could still form tubes if apical constriction was blocked, indicating that it is not an essential part of tissue invagination in this organ. However, blocking apical constriction led the tube to develop unusual shapes at intermediate stages.

More work is now needed to better understand the links between apical constriction, cell rearrangement and tissue invagination. These processes are fundamental for organs to form correctly in many organisms and understanding their control could have wide-ranging impacts. A better understanding of these processes may provide insight into how the tubes can form while keeping all the cells adequately supplied with oxygen and nutrients, and into diseases that result if there are defects in the invagination process.

the Folded gastrulation (Fog) pathway that regulates apical constriction and apicomedial myosin formation (*Manning and Rogers, 2014*). During gastrulation, mesodermal cells undergo apical constriction to form the ventral furrow along the anterior/posterior body axis. In those cells, the mesoderm-specific transcription factors Twist and Snail activate G protein-coupled receptor signaling and recruit RhoGEF2 to the apical surface, which, in turn, activates Rho1 (*Costa et al., 1994*; *Kölsch et al., 2007*; *Manning et al., 2013*; *Parks and Wieschaus, 1991*). GTP-bound Rho1 then activates Rho-associated kinase (Rok), which phosphorylates and activates non-muscle myosin II, which forms an actomyosin complex at the medial apical cortex (*Dawes-Hoang et al., 2005*). This actomyosin complex causes asynchronous contractions that pull the adherens junctions (AJs) inward. Contractions are maintained between pulses by the actomyosin belt, which serves as a 'ratchet' to incrementally reduce apical area (*Martin et al., 2009*).

Although apical constriction and its associated forces are suggested to drive tissue invagination, the exact role of this cell shape change in tube formation remains controversial (*Llimargas and Casanova, 2010*). In *Drosophila* trachea defective for EGF receptor signaling, apical constriction is impaired, but most cells invaginate (*Brodu and Casanova, 2006*; *Nishimura et al., 2007*). Similarly, in *Drosophila* embryos mutant for *twist* or *fog*, mesodermal cells with defective apical constriction still invaginate, although the process is both delayed and aberrant (*Leptin and Grunewald, 1990*; *Sweeton et al., 1991*). In these mutants, however, apical constriction is not completely blocked; it is simply less extensive and more random (*Brodu and Casanova, 2006*; *Costa et al., 1994*; *Nishimura et al., 2007*; *Sweeton et al., 1991*), making it difficult to draw any clear conclusions. A recent study, using an optogenetic method to deplete phosphatidylinositol-4,5 bisphosphate (PI(4,5) P$_2$) and actin from the cell cortex, showed that local inhibition of apical constriction is sufficient to cause global arrest of mesoderm invagination during *Drosophila* gastrulation (*Guglielmi et al.,*

*2015*). This finding suggests that apical constriction is essential for the invagination by wrapping that occurs during ventral furrow formation. It remains unclear, however, whether apical constriction is also critical for tissue invagination by budding.

The *Drosophila* salivary gland (SG) is an excellent system to study the role of apical constriction during tissue invagination by budding (*Figure 1A–A'',B,B',C and C'*). The SG begins as a two-dimensional sheet of cells on the embryo surface that internalizes to form an elongated tube (*Chung et al., 2014*). Since neither cell division nor cell death occurs once the SG has been specified, the entire morphogenetic process must be driven by changes in cell shape and rearrangement. Indeed, apical constriction has been observed in this tissue (*Myat and Andrew, 2000a*), and an increase in apical myosin has been reported during SG invagination (*Escudero et al., 2007*; *Nikolaidou and Barrett, 2004*; *Xu et al., 2008*). More detailed analyses revealed several distinct myosin structures in the forming SG, including a supracellular myosin cable that surrounds the entire tissue and is thought to be involved in tissue invagination, as well as a web-like myosin structure in the apicomedial region of cells that colocalizes with actin (*Röper, 2012*). The latter shows pulsatile behavior, suggesting that contractile forces by the apicomedial actomyosin complex may drive apical constriction during SG invagination (*Booth et al., 2014*).

Here, we elucidate the mechanism by which apical constriction is regulated during SG budding and determine its role in tissue invagination. We show that the spatial and temporal pattern of apical constriction correlates with apicomedial myosin formation during SG invagination. We uncover the molecular pathway through which the FoxA transcription factor Fork head (Fkh) coordinates apical constriction by regulating Fog signaling in the SGs. Through genetic manipulations to completely block apical constriction and detailed quantitative analysis, we show that apical constriction is not required for SG invagination, but is required for proper tissue geometry. SG cells can internalize even without forming apicomedial myosin, suggesting a role for other additional forces. We provide evidence that the compressing force generated by the tissue-level supracellular myosin cable contributes to invagination in SGs with defective apical constriction. By analyzing the externalized SG phenotypes of *fog* mutants, we also reveal that tube formation can be decoupled from tissue invagination.

## Results

### SG invagination involves coordinated apical constriction and requires fork head

To gain a detailed view of temporal and spatial regulation of apical constriction during SG invagination, we analyzed apical area in wild type (WT) SGs of stage 11 embryos. Using labeling with E-Cadherin (E-Cad), an AJ marker, and CrebA, an SG nuclear marker, we segmented apical cell outlines and calculated the apical area of individual cells. By analyzing >60 WT SGs for apical area and depth of invagination, we classified four distinct stages (*Figure 1D–G, D'–G'*). At early stage 11, all SG cells were on the embryo surface and cells with different apical area were distributed relatively randomly, although cells with small and large apical area tended to be toward the inside and at the periphery of the tissue, respectively (Before invagination; *Figure 1D and D'*). Shortly after, subsets of cells clustered in specific regions had smaller apical areas, most notably in a middle region close to the posterior edge of the SG where invagination begins (Clustered apical constriction; *Figure 1E and E'*). The less prominent anterior cluster corresponds to a small indentation that can be observed in lateral views of stage 12 SGs (data not shown). When a few cells in the posterior cluster had internalized to form an invagination pit, many cells near the pit were apically constricted (Beginning of invagination; *Figure 1F and F'*). At late stage 11, as more cells had invaginated, most cells anterior to the invagination pit were apically constricted (Deep invagination; *Figure 1G and G'*). At the same time, cells in the periphery of the SG were apically expanded, especially in the region ventral to the invagination pit. Quantification of the percentage and cumulative percentage of cells of different apical area showed a gradual and significant increase in the number of cells with smaller apical area as invagination progressed. Small increases in the number of cells with larger apical area were also observed (*Figure 1I and J*).

The SGs of *fkh* null mutants undergo early apoptosis and fail to internalize (*Myat and Andrew, 2000a*). *fkh* mutant SGs rescued from cell death by deletion of the pro-apoptotic genes contained

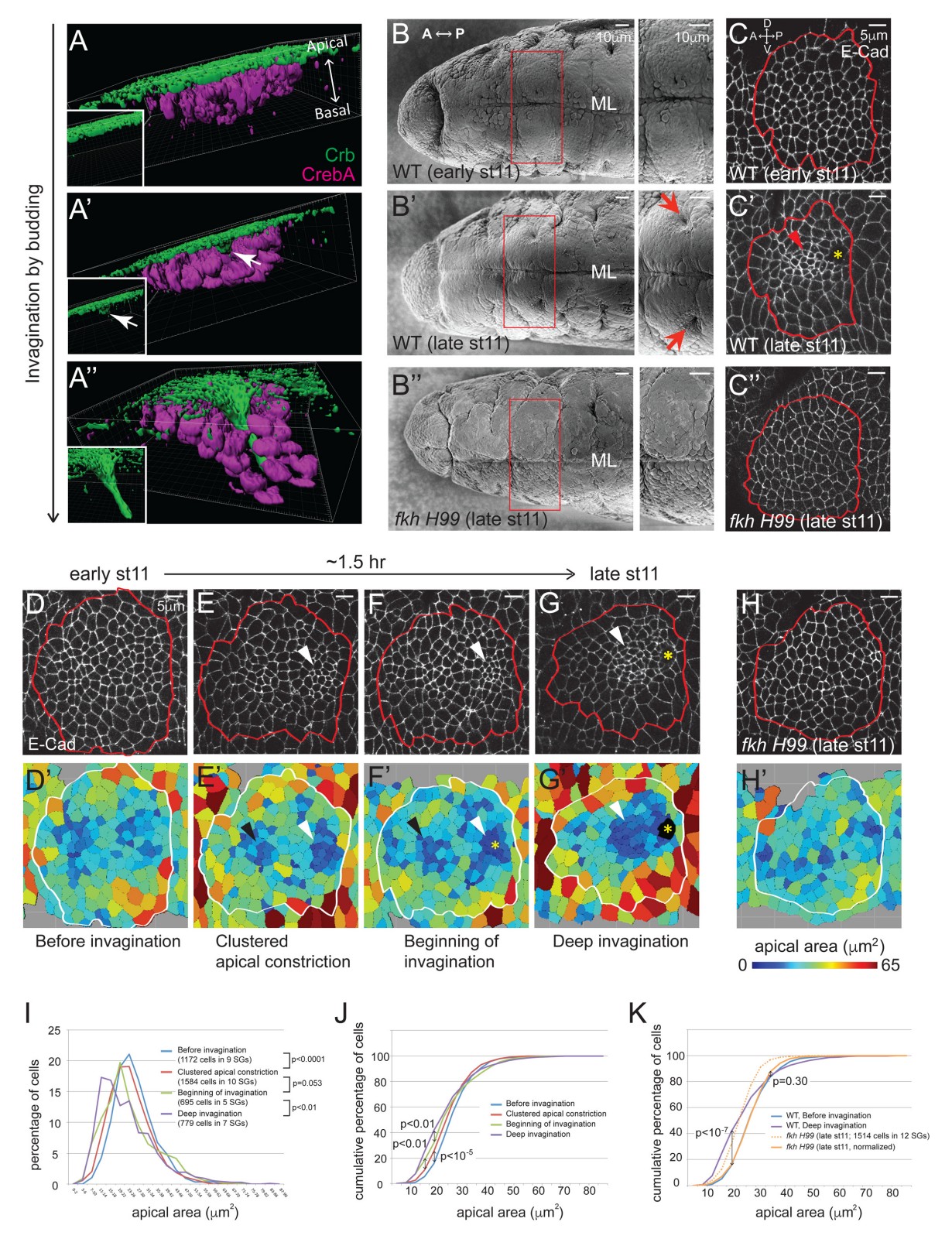

**Figure 1.** Clustered apical constriction does not occur in *fkh* mutant SGs. (**A–A''**) Epithelial invagination by budding. 3D reconstruction of *Drosophila* embryonic SGs stained with Crb (green), an apically localized transmembrane protein, and CrebA (magenta), an SG nuclear transcription factor, before (**A**), at the beginning (**A'**) and during (**A''**) invagination. White arrows in **A'** indicate the budding epithelium. Insets, Crb signals only. (**B**) SEM images of ventral views of early and late stage 11 embryos show two SG placodes, with higher magnification to the right. An invagination pit is observed in the

*Figure 1 continued on next page*

Figure 1 continued

posterior dorsal region of each placode at late stage 11 in WT (arrows in **B'**), which is absent in *fkh H99* mutants (**B''**). ML, midline. (**C**) E-Cad staining of WT and *fkh H99* mutant SGs of early and late stage 11. Ventral views of a single SG placode are shown. Red lines denote the border of the SG placodes, based on a CrebA staining (not shown). In this figure and later, anterior is to the left and dorsal is up. Robust apical constriction is observed in WT SGs of late stage 11 (arrowhead in **C'**), which is not detected in WT SG of early stage 11 (**C**) or in *fkh H99* SG of late stage 11 (**C''**). Asterisk, invagination pit. (**D–H**) Coordinated apical constriction is observed in WT SGs prior to and during invagination, which does not occur in *fkh* mutants. Representative SGs for the four distinct stages of invagination observed in WT (**D–G**) and late stage 11 *fkh H99* embryos (**H**) and the corresponding heat maps of apical area are shown (**D'–H'**). Apical area of each cell was calculated by automated tracing of E-Cad along cell boundaries. Red and white lines denote the border of the SG placodes. Arrowheads, clustered apical constriction in the posterior (white) or anterior (black) region of the placode. Asterisks, invagination pit. (**I–K**) Percentage (**I**) and cumulative percentage (**J**) of WT SG cells in different apical area bins at each stage of invagination. Comparison of the cumulative percentage of cells in WT and *fkh* mutants (**K**). P values are calculated using the Mann-Whitney U-test (**I**) and the Kolmogorov-Smirnov test (**J, K**). See also *Figure 1—source data 1*.

The following source data is available for figure 1:

**Source data 1.** SG cells quantified for apical area.

within the *H99* deficiency also fail to internalize. SGs of *H99* deficiency embryos develop completely normally during embryogenesis, indicating that blocking apoptosis does not affect SG morphogenesis (*Myat and Andrew, 2000a*). Scanning electron microscope (SEM) and confocal images of the late stage 11 *fkh H99* SGs did not reveal invagination pits (*Figure 1B'' and C''*), consistent with a complete failure of invagination. Analysis of apical areas of late stage 11 *fkh H99* SGs revealed that *fkh H99* SG cells had overall smaller apical area than WT cells; both the mean and median values for apical area were lower than those of WT (*Figure 1—source data 1*). Importantly, cells with the smallest apical areas were not clustered in *fkh H99* mutant SGs and were instead randomly distributed (*Figure 1H and H'*). Moreover, the cumulative percentage of late stage 11 *fkh H99* SG cells of different apical areas showed a similar distribution trend to that of WT SG cells before invagination (*Figure 1K*). Indeed, median scaling normalization of the apical area of *fkh H99* late stage 11 SG cells revealed a very similar distribution to the apical area of WT SGs before invagination. Particularly, the percentages of cells with the lower 70% apical area were indistinguishable (*Figure 1K*). Taken together, our analysis reveals that apical constriction is both temporally and spatially coordinated during SG invagination and does not occur in *fkh H99* mutants.

## *fkh* is required for apicomedial myosin accumulation

Apical constriction of both vertebrate and invertebrate tissues is associated with contraction of the actomyosin cytoskeleton (*Dawes-Hoang et al., 2005*; *Martin et al., 2009*; *Nishimura et al., 2007*; *Roh-Johnson et al., 2012*). Specifically, pulsatile accumulation of apicomedial myosin has been linked to apical constriction (*Martin et al., 2009*; *Blanchard et al., 2010*; *Booth et al., 2014*). Therefore, we investigated myosin accumulation in WT and *fkh H99* SGs, using *spaghetti squash* (*sqh*)-GFP, a functional tagged version of the myosin regulatory light chain (*Royou et al., 2004*). Consistent with results from (*Röper, 2012*), several myosin structures were observed, including a supracellular myosin cable that encircles the whole tissue, AJ-associated cortical myosin, as well as an apicomedial myosin web (*Figure 2A–C*). Importantly, apicomedial myosin formed in a manner that recapitulates the observed temporal and spatial pattern of apical constriction. Before invagination, myosin predominantly localized at the cortical regions in all SG cells, partially colocalizing with E-Cad and often showing higher intensity at vertices (hereafter referred to as junctional myosin; *Figure 2A*). When invagination began, moderate levels of apicomedial myosin were observed in cells near where invagination initiates (*Figure 2B*). During deep invagination, apicomedial myosin was quite prominent in the cells anterior to the invagination pit (*Figure 2C*). Indeed, the measured intensity ratios of junctional to apicomedial myosin decreased over time in the WT SG cells (*Figure 2E*). In contrast, junctional myosin was predominant throughout stage 11 in *fkh H99* SGs, and only dispersed, weak myosin signals were observed in the apical region (*Figure 2D*). Correspondingly, the ratio of junctional to apicomedial myosin in *fkh H99* SGs was comparable to that of WT before invagination (*Figure 2E*).

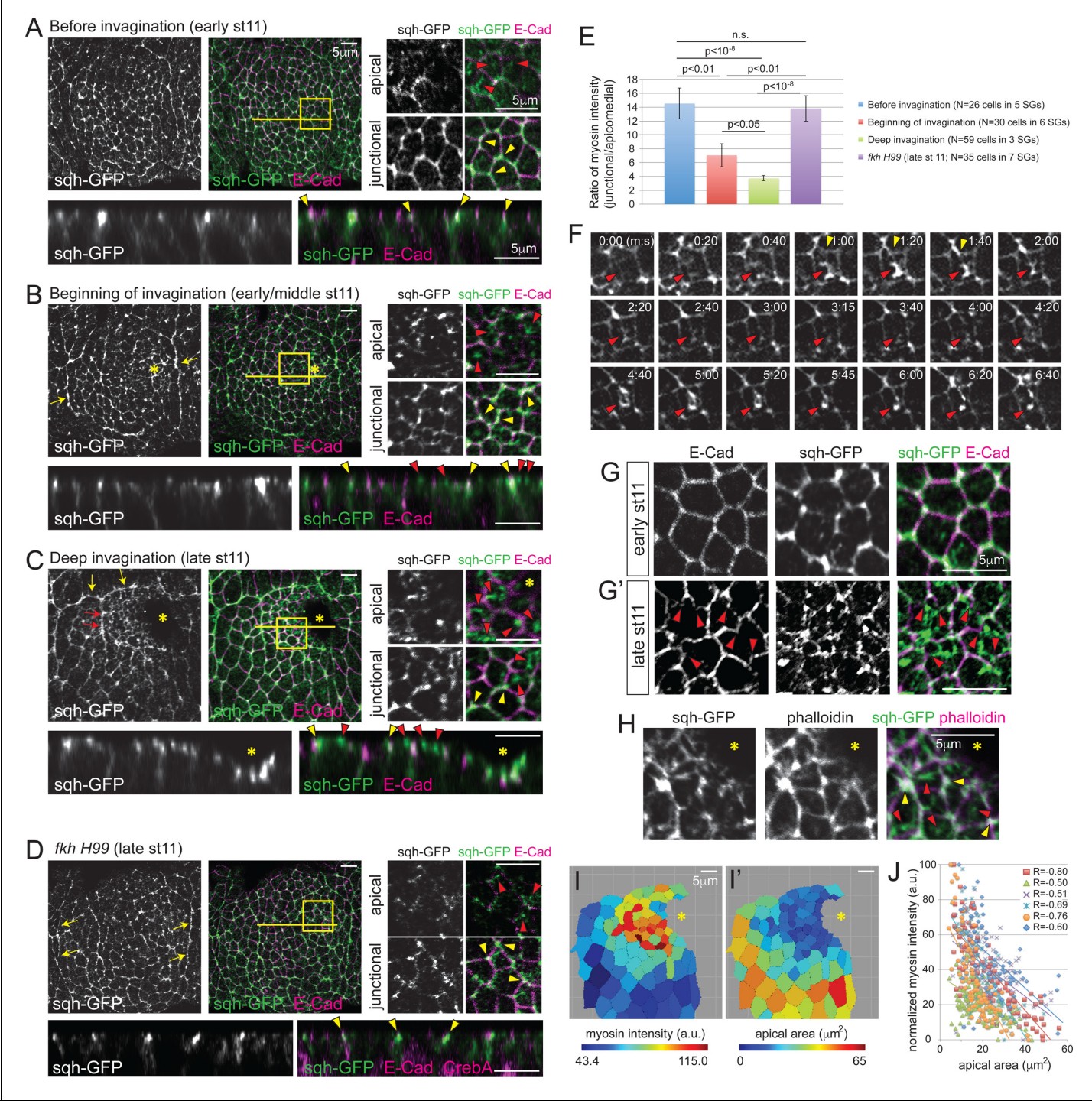

**Figure 2.** Apicomedial myosin accumulation and coordinated apical constriction are regulated both spatially and temporally during SG invagination. (A–D) Myosin accumulation at different stages of invagination of WT (A–C) and in the *fkh* mutant SGs (D). Two focal planes (apical and junctional) for the posterior/dorsal region of each placode (yellow box) are shown in higher magnification. Z sections along the yellow lines are shown at the bottom. Before invagination (A), most myosin is found along cell junctions, often with higher intensity at vertices (yellow arrowheads). Only very weak and dispersed myosin is observed apically (red arrowheads). When the first cells begin to invaginate (B), web-like myosin structures become prominent in the apical region of cells (red arrowheads). Strong myosin signals at cell junctions are still observed (yellow arrowheads). Supracellular myosin cables along the anterior and posterior boundaries of the SG are also observed (yellow arrows). During deep invagination (C), high intensity apicomedial myosin web structures are observed in cells near the invagination pit (red arrowheads). Note that the epithelial sheet is tilted a little basally toward the invagination pit (asterisk) in the magnified images. At this stage, large supracellular cables form at the dorsal boundary of the tissue (yellow arrows) to

*Figure 2 continued on next page*

*Figure 2 continued*

connect the lateral cables and surround the entire tissue. Short intercellular myosin cables across several cells are also occasionally observed (red arrows). In *fkh* mutant SGs (D), strong myosin signals are observed only along junctions, even at late stage 11 (yellow arrowheads). Apical myosin is weak and dispersed (red arrowheads). The supracellular myosin cables along the lateral boundaries of the tissue are still visible (arrows), but a connected dorsal cable does not form. (E) Ratio of junctional to apicomedial myosin signals of the SG cells. Shown are mean ± SEM. P values are calculated using the two-tailed Student's t-test. See *Figure 2—source data 1*. (F) Time-lapse images of sqh-GFP in a single WT SG cell show pulsatile behavior of apicomedial myosin (red arrowheads). Cell deformation is occasionally observed during the peak intensity period of apicomedial myosin (yellow arrowheads). See *Video 1*. (G) Whereas cells have a roughly hexagonal shape before invagination (G), significant cell membrane distortion is observed during invagination (G') where apicomedial myosin contacts E-Cad (arrowheads). (H) Weak actin signals colocalize with apicomedial myosin (red arrowheads). Strong actin signals colocalize with myosin at cell junctions (yellow arrowheads). (I) Heat maps corresponding to the late stage 11 WT SG shown in (C). Higher intensity myosin signals are observed in cells near the invagination pit (I), which have smaller apical area (I'). (J) Negative correlation between myosin intensity and apical area during deep invagination. Myosin intensity for each SG is re-scaled for 0 to 100 (a.u.). Cells from six WT SGs are plotted with different colors. Trendlines are shown for each SG. R, Pearson correlation coefficient. p<0.0001 for all samples. Asterisks in B, C, H and I, invagination pit.

The following source data is available for figure 2:

**Source data 1.** Ratio of junctional to apicomedial myosin signals of the SG cells.

## Pulsatile apicomedial myosin correlates with SG cell deformation and apical constriction

To investigate the dynamics of apicomedial myosin in the SG, we took time-lapse images of sqh-GFP. Apicomedial myosin in the SG was also pulsatile, with an average interval of 131.7s ± 42.8 s between pulses (mean ± s.d., n = 14 cells in 4 SGs, 20 pulses; *Figure 2F*; *Video 1*). Pulses were mostly asynchronous between adjacent cells. The apicomedial myosin colocalized with F-actin, indicating formation of an actomyosin complex (*Figure 2H*; *Röper, 2012*). Importantly, we observed cell distortions during the peak intensity of pulses where the apicomedial myosin contacts E-Cad (*Figure 2F and G'*), reminiscent of the transient bending of AJs in constricting mesodermal cells (*Martin et al., 2009*), suggesting that actomyosin contraction generates a pulling force during SG invagination. Measurements of average myosin intensity in the apical domain of each cell showed a negative correlation with the apical area of cells in the entire SG (*Figure 2I, I' and J*). This correlation, combined with the relative increase in apicomedial myosin in cells anterior to the invagination pit, suggests a role for this myosin pool in the clustered apical constriction observed during SG invagination.

## Fkh acts through apicomedial Rok accumulation

Rho kinase (Rok) is a key regulator for myosin phosphorylation and activation (*Amano et al., 1996*). During ventral furrow formation, Rok is polarized to an apicomedial domain of mesodermal cells, where it promotes assembly of apicomedial myosin (*Mason et al., 2013*). Therefore, we analyzed Rok localization dynamics using a ubiquitously expressed GFP-tagged Rok transgene (Rok-GFP; *Abreu-Blanco et al., 2014*). Rok localization recapitulated the temporal and spatial distribution of apicomedial myosin. Rok was initially detected as occasional small punctate structures in the apical domain of all SG cells (*Figure 3A*), but as invagination proceeded, gradually larger apical puncta were observed in the posterior region of the SG (*Figure 3B*). During deep invagination, huge globular structures of Rok occupied nearly the entire apicomedial domain (*Figure 3C*). Live imaging revealed congregating and separating behaviors, suggesting

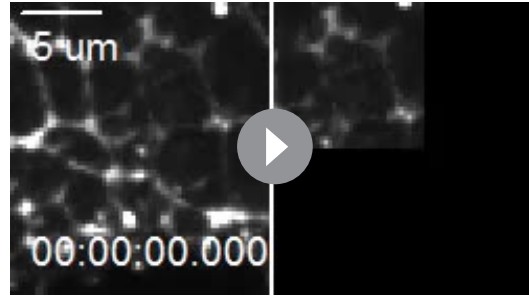

**Video 1.** Pulsatile apicomedial myosin, Related to *Figure 2*. Time-lapse movie of sqh-GFP in a WT SG during invagination; a single confocal section is shown. Signals in a single SG cell are shown in the right panel. Frames are 5 s apart.

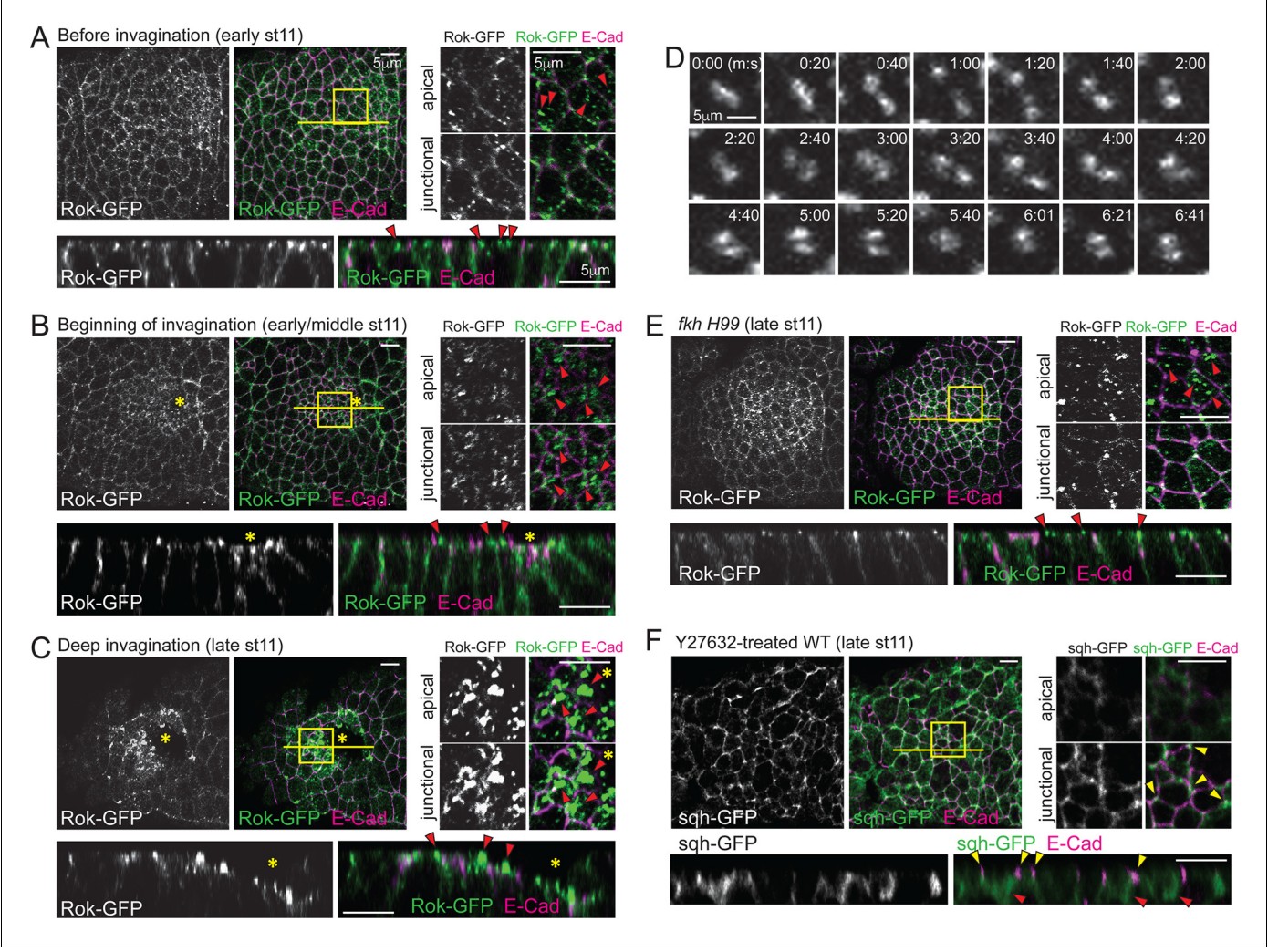

**Figure 3.** Spatiotemporal regulation of Rok is critical for formation of apicomedial myosin during SG invagination. (A–C) Apicomedial Rok increases dramatically during SG invagination and forms huge globular structures. Two focal planes (apical and junctional) for the posterior/dorsal region of each placode (yellow box) are shown at higher magnification. Bottom panels for each time point are the Z sections along the yellow lines. Before invagination (A), Rok is observed only as small puncta in the apical region (arrowheads). Additional Rok signals are shown along the entire lateral membranes. When cells first begin to invaginate (B), Rok is observed in large punctate structures in the apical region of the posterior/dorsal region of the placode (arrowheads). During deep invagination (C), huge globs of Rok accumulation are observed in cells near the invagination pit (arrowheads). Asterisk, invagination pit. (D) Time-lapse images of Rok-GFP in a single WT SG cell show dynamic apicomedial Rok accumulation. See *Video 2*. (E) In late stage 11 *fkh* mutants, apical Rok is present only in small punctate structures (arrowheads). (F) Y-27632 inhibits formation of apicomedial myosin and SG invagination. Myosin is only observed along the lateral membrane, including the AJ domain (yellow arrowheads) and in the basal region of the cells (red arrowheads).

that the large globular structures are dynamic (*Figure 3D*; *Video 2*). Importantly, only small punctate signals were detected in the apical region of *fkh H99* SG cells throughout all of stage 11 (*Figure 3E*), indicating that Fkh is required for apicomedial Rok accumulation. In embryos treated with the Rok inhibitor Y-27632, myosin was detected only along the lateral membrane and in the basal region of the cells, suggesting that Rok activity is required for the formation of apicomedial myosin (*Figure 3F*). Furthermore, the SGs of Y-27632-treated embryos did not invaginate, suggesting a critical role for Rok.

## Fog coordinates apical constriction by controlling apicomedial accumulation of myosin and Rok

We next asked which effector molecule acts downstream of Fkh to regulate apicomedial accumulation of Rok and myosin in SG cells. A good candidate was Fog, a secreted ligand known to regulate apical constriction of cells in several tissues (*Costa et al., 1994*; *Dawes-Hoang et al., 2005*; *Nikolaidou and Barrett, 2004*). Fluorescent in situ hybridization experiments revealed that *fog* mRNA is upregulated in the SG prior to and during invagination (*Figure 4A*; *Nikolaidou and Barrett, 2004*) and persists until at least stage 14 (data not shown). Importantly, SG expression of *fog* requires Fkh; only background levels of *fog* mRNA were observed in *fkh H99* SGs (*Figure 4A'*).

Unlike *fkh* mutants, *fog* mutant SGs invaginate. *fog* mutant SGs formed an invagination pit at a relatively normal position at about the same stage as WT, although the pit was often somewhat larger (*Figure 4C*). Cell segmentation analyses revealed that the percentage and cumulative percentage of cells of different apical area did not show significant differences between *fog* mutant and WT SGs (*Figure 4D*; *Figure 4—source data 3*). Cells with smaller apical area, however, showed a less coordinated spatial distribution in *fog* mutants. Unlike in WT SGs, where apically constricted cells were tightly clustered anterior to the invagination pit (*Figure 4B and B'*), in *fog* mutant SGs, apically constricted cells were dispersed, with significantly increased dispersion along the dorsal/ventral (D/V) axis of the tissue (*Figure 4C, C', E and F*).

Consistent with the uncoordinated apical constriction in *fog* mutant SGs, analysis of myosin signals revealed less organized web-like structures with lower staining intensity than with age-matched WT samples (*Figure 4G*; compare to *Figure 2C*). The ratio of intensity between junctional and apicomedial myosin in *fog* mutants during deep invagination was more like that of WT glands at earlier stages - between the value measured at the before invagination and at the beginning of invagination stages (*Figure 4I*; compare to *Figure 2E*).

We then asked if the apicomedial accumulation of Rok is affected by loss of *fog*. Although apicomedial Rok has been reported during ventral furrow formation (*Mason et al., 2013*), its dependence on *fog* has not been addressed. In *fog* mutant SGs, we observed Rok in small punctate structures dispersed along the apical domain during deep invagination, rather than in the huge globular structures observed in WT SG cells at this stage (*Figure 4H*; compare to *Figure 3C*). These data indicate that reduced Rok accumulation in the apicomedial region of *fog* mutant SG cells leads to reduced apicomedial myosin.

## Blocking apical constriction does not arrest SG invagination

Since the loss of *fog* only affected the pattern of apical constriction, we next sought to completely block apical constriction. Overexpression of Crb, which expands the apical domain when overexpressed (*Wodarz et al., 1995*), or a constitutively-active form of Diaphanous (Dia-CA; *Somogyi and Rørth, 2004*), a fly formin protein that nucleates and facilitates the elongation of actin filaments (*Higgs and Peterson, 2005*), from the onset of SG specification throughout development efficiently blocked apical constriction; clustered apical constriction was not observed at any stage of invagination (*Figure 5A–H, A'–H'*). Moreover, analyses of the percentage and cumulative percentage of cells of different apical area showed that Crb or Dia-CA overexpression not only blocked apical constriction but also caused significant increase in apical area. Before invagination, Crb-overexpressing cells showed apical areas comparable to those of WT cells at early stage 11, and Dia-CA-overexpressing cells showed a small (but significant) increase of apical area (*Figure 5I*; *Figure 5—source data 1*). At the beginning of invagination, cells showed a notable increase in apical area, with obvious shifts in the distribution of apical area when compared to WT (*Figure 5I*; *Figure 5—source data 1*). Importantly, the percentage of cells in different apical area bins of Crb- or Dia-CA-overexpressing SGs was comparable to that of WT SG cells before invagination, confirming that the cells do not undergo apical constriction (*Figure 5I*; *Figure 5—source data 1*). During deep invagination, Dia-CA-overexpressing SG cells still have apical areas comparable to those of WT SG cells before invagination, but overexpression of Crb caused a huge increase in apical area (*Figure 5I*; *Figure 5—source data 1*). Importantly, however, neither blocking apical constriction nor increasing apical area arrested SG invagination. SGs overexpressing Crb or Dia-CA formed invagination pits without any delays. Plotting of cells of lower 10%, 20% and 30% of apical area showed that cells with smaller apical area were not as clustered as in WT, with significantly increased dispersion along the anterior/

posterior (A/P) axis of the tissue (*Figure 5—figure supplement 1A and B*). These data indicate that overexpression of Crb or Dia-CA effectively blocks apical constriction and alters the distribution of cells with smaller apical areas.

To understand how apical constriction is blocked in Crb- and Dia-CA-overexpressing SGs, we asked if the cellular machinery required for apical constriction was affected. Even during deep invagination, Rok signals were detected only as small- to medium-sized puncta (in Crb-overexpressing SG cells; *Figure 5J*) or as very small puncta (in Dia-CA-overexpressing SG cells; *Figure 5L*) rather than the huge globular structures detected in WT cells (*Figure 3C*). However, Crb-overexpressing SGs formed relatively normal pools of apicomedial myosin that are associated with actin and with membrane distortions (*Figure 5K*), consistent with our findings that in *fog* mutants even dispersed Rok can still result in some apicomedial myosin formation (*Figure 4F and G*). These data suggest that in Crb-overexpressing SG cells, the forces of apical expansion overwhelm those of apical constriction despite the actomyosin machinery being in place and functional. Interestingly, in Dia-CA-overexpressing SG cells, most myosin signals were observed in a broad cortical area partially overlapping with actin, but no web-like apicomedial myosin formed (*Figure 5M*; compare to *Figure 2C*). A similar increase in cortical myosin was observed in Dia-CA-overexpressing amnioserosa cells, but interestingly, premature apical constriction was observed in those cells (*Homem and Peifer, 2008*), suggesting possible tissue-specific effects. Moreover, the significant 'wiggliness' of the AJs in Dia-CA-overexpressing SGs suggests a decrease of cortical tension (*Figure 5E, G, H and L*; *Blanchard et al., 2010*; *Choi et al., 2016*; *Lecuit and Lenne, 2007*; *Martin et al., 2009*). Overall, since SG invagination occurs despite the failure in apical constriction with overexpression of either Crb or Dia-CA, we conclude that apical constriction is not required.

## Apical constriction is required for the correct geometry of tissue invagination

Although invagination pits formed at approximately the right position and at the right time in the SGs with either uncoordinated (in *fog* mutants) or completely blocked apical constriction (in Crb- or Dia-CA-overexpressing SGs), the pits were abnormal. Specifically, both the *fog* mutant and Crb-overexpressing SGs frequently formed a pit that was elongated along the D/V axis, rather than the smaller oval-shaped pit of WT SGs (*Figure 6A–C, A′–C′*). Overexpression of Dia-CA resulted in a huge invagination pit, often larger than half the size of the tissue (*Figure 6D and D′*). We asked how this aberrant invagination and the continuous increases in apical area affected tube structure. 3D reconstruction of invaginating SGs (late stage 11) revealed a wider lumen in *fog* mutant and Crb- and Dia-CA-overexpressing SGs compared to WT, indicating that early tube architecture is affected by defective apical constriction and consequent abnormal invagination (*Figure 6E–6H*). We next analyzed stage 14 embryos, when in WT, all SG cells have fully internalized (*Figure 6I*). Interestingly, Dia-CA-overexpressing SGs showed a significant delay in cell internalization and migration. All Dia-CA-overexpressing cells eventually internalized, but formed a short gland with increased luminal diameter that remained close to the embryo surface (*Figure 6J*). Overproduced Crb not only localized to the apical membranes, but also mislocalized to the basal membranes facing outside of the tube; nonetheless, SG cells still formed a single-layered fully internalized tube at stage 14 (*Figure 6K*). Although Crb overexpression did not delay SG

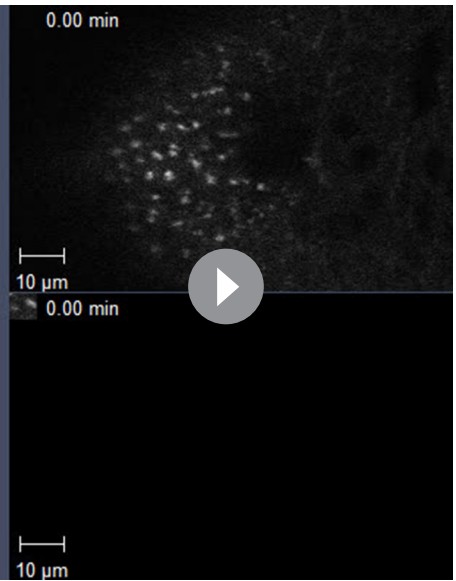

**Video 2.** Rok is observed as huge globular structures in the invaginating SGs, Related to *Figure 3*. Time-lapse movie of Rok-GFP in a WT SG during invagination; a single confocal section is shown. Signals in a single SG cell are shown in the bottom panel. Frames are 20 s apart.

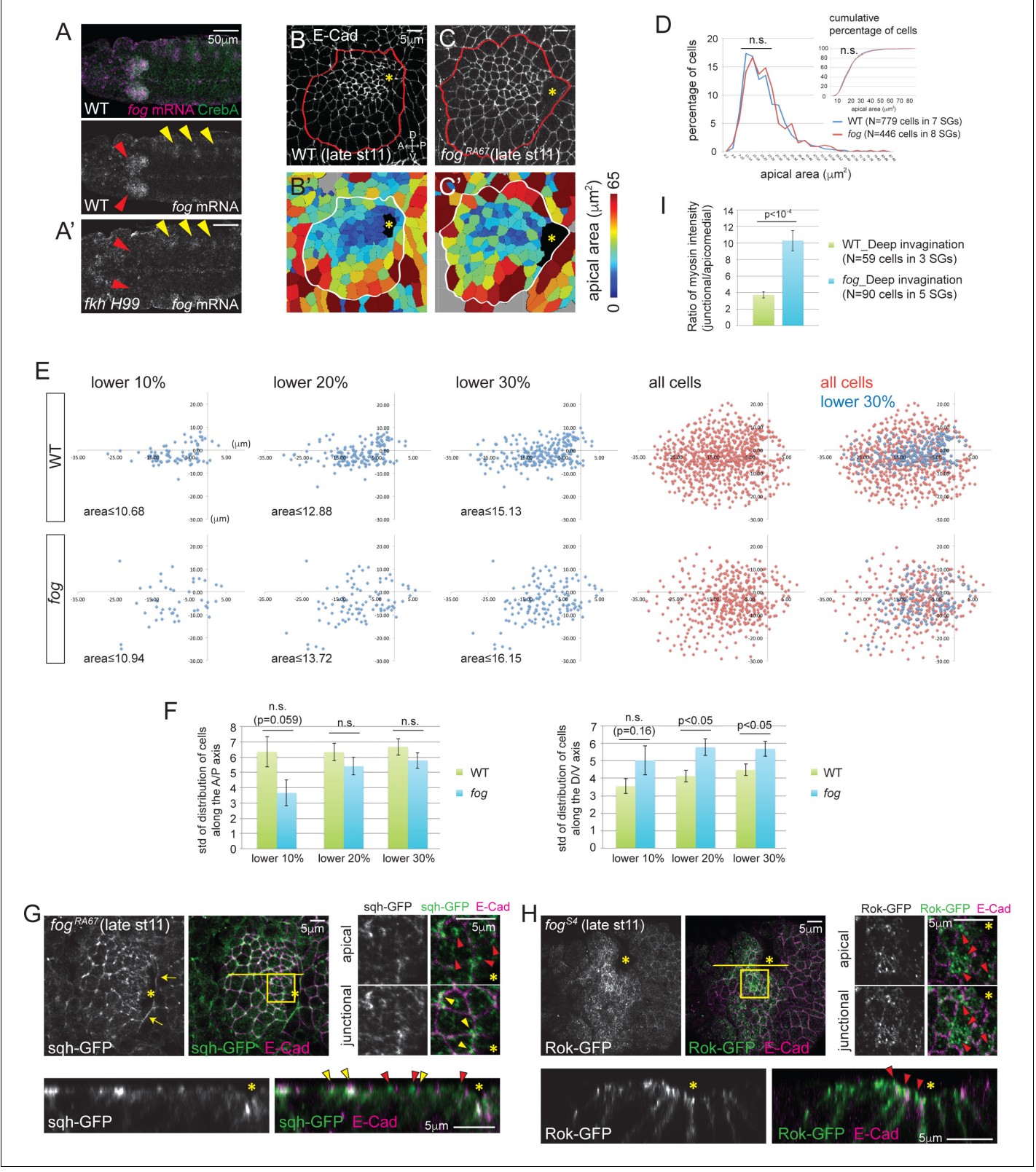

**Figure 4.** Fog, a downstream effector of Fkh, is essential for proper Rok localization, apicomedial myosin formation and coordinated apical constriction. (A) *fog* mRNA (magenta) is expressed in the SG (red arrowheads in **A**), overlapping with the SG-specific marker CrebA (green). In *fkh* mutants, *fog* mRNA in the SG is at background levels (red arrowheads in **A'**) whereas *fog* expression in the developing trachea is unaffected (yellow arrowheads). (**B, C**) E-Cad staining of WT and *fog* SGs during invagination (**B, C**) and the corresponding heat maps of apical area (**B', C'**). (D) Percentage and cumulative

*Figure 4 continued on next page*

*Figure 4 continued*

percentage of cells of different apical area in WT and *fog* mutant SGs are indistinguishable during invagination. See *Figure 4—source data 1*. (E) Scatter plots showing the position of cells relative to the invagination pit. X and Y axes represent the distance along the A/P axis and D/V axis from the pit, respectively. Cells of lower 10%, 20%, 30% of apical area (blue) and all cells (red) are plotted. Note that the cells of lower 30% of area are plotted on top of all cells in the merged plots (right-most panels). The same cells quantified in D were analyzed in E and F. (F) Quantification of distribution of cells along the A/P and D/V axis. Compared to WT, cells with small apical area are more dispersed along the D/V axis in *fog* mutant SGs. See *Figure 4—source data 2*. (G) Only weak apicomedial myosin structures form in *fog* mutant SGs during invagination (red arrowheads). Yellow arrowheads, junctional myosin. (H) Rok is more dispersed at the apical region and is observed only in small punctate structures (arrowheads) in *fog* mutant SGs. (I) The ratio of junctional to apicomedial myosin is significantly higher in *fog* mutants during deep invagination. Shown are mean ± SEM. P values are calculated using the two-tailed Student's t-test. See *Figure 4—source data 3*. Asterisks in B, C, G and H, invagination pit.

The following source data is available for figure 4:

**Source data 1.** SG cells quantified for apical area.
**Source data 2.** Distribution of SG cells along the A/P and D/V axis.
**Source data 3.** Ratio of junctional to apicomedial myosin signals of the SG cells.

internalization or migration, partially or completely externalized SGs with no clear internal lumen were occasionally observed at stage 15 and 16 (*Figure 6L*). Since all SGs were internalized at earlier stages, these findings suggest that the Crb-overexpressing SG cells subsequently evaginated from the inside to the outside of the embryo, likely because of continued apical expansion. Similar invagination-followed-by-evagination phenotypes were observed when a myristoylated, membrane-tethered form of Wiskott–Aldrich syndrome protein (Myr-Wasp), an activator of the actin nucleating Arp2/3 complex, was overexpressed in the SGs (*Figure 6M*; *Video 3*). Overall, these data suggest that the major role of apical constriction is to ensure the proper tissue geometry during SG invagination.

## The supracellular myosin cable contributes to SG invagination

The supracellular myosin cable that surrounds the entire SG is under tension (*Röper, 2012*), suggesting that it generates a compression force. Therefore, we asked if this tissue-scale myosin cable is present in the SGs defective for apical constriction and could contribute to invagination. At late stage 11, *fog* and Crb-overexpressing SGs formed a supracellular myosin cable almost indistinguishable from that of WT (*Figure 7A–, A'–C'*). However, this cable was defective in both Dia-CA-overexpressing SGs and in *fkh H99* mutants, where invagination is either delayed or completely fails, respectively. In Dia-CA-overexpressing SGs, myosin signals were observed in a broad cortical area of individual cells (*Figure 5M*) and no obvious cable structure surrounding the tissue was observed (*Figure 7D and D'*). In *fkh H99* mutants, myosin cables were observed along the lateral boundaries of the tissue, coinciding with the parasegmental boundaries, but the connecting dorsal cable was not observed (*Figure 7E and E'*). Similar myosin cables are observed in the lateral boundaries of *Scr* mutants (*Röper, 2012*), which do not form SGs (*Andrew et al., 1994*).

We calculated the circularity of the SG placode as a measure of smoothness and tension (See Materials and methods for details). The circularity of the WT placode is slightly, but significantly increased from early stage 11 to late stage 11, suggesting an increased tension over time (in *Figure 7F*). At late stage 11, the border of the WT placode where the supracellular cable was observed was very smooth, and circularity of the placode boundary was significantly greater than that of the boundaries shifted outside the placode by one cell (outer boundary) or inside the placode by one cell (inner boundary), suggesting that this big cable is under tension (*Figure 7F and G*). The circularity of the *fog* and Crb-overexpressing SG placode was comparable to that of WT, and was significantly greater than that of the outer and inner boundaries (*Figure 7F*). Consistent with the lack of a supracellular cable, the circularity of the Dia-CA-overexpressing SG boundary was significantly lower than that of WT, and was not significantly different from the circularity of the inner boundary, suggesting disrupted tension at the SG boundary (*Figure 7F*). The circularity of the *fkh H99* mutant SG boundary was between that of early and late stage 11 WT, and was also comparable to the *fkh*

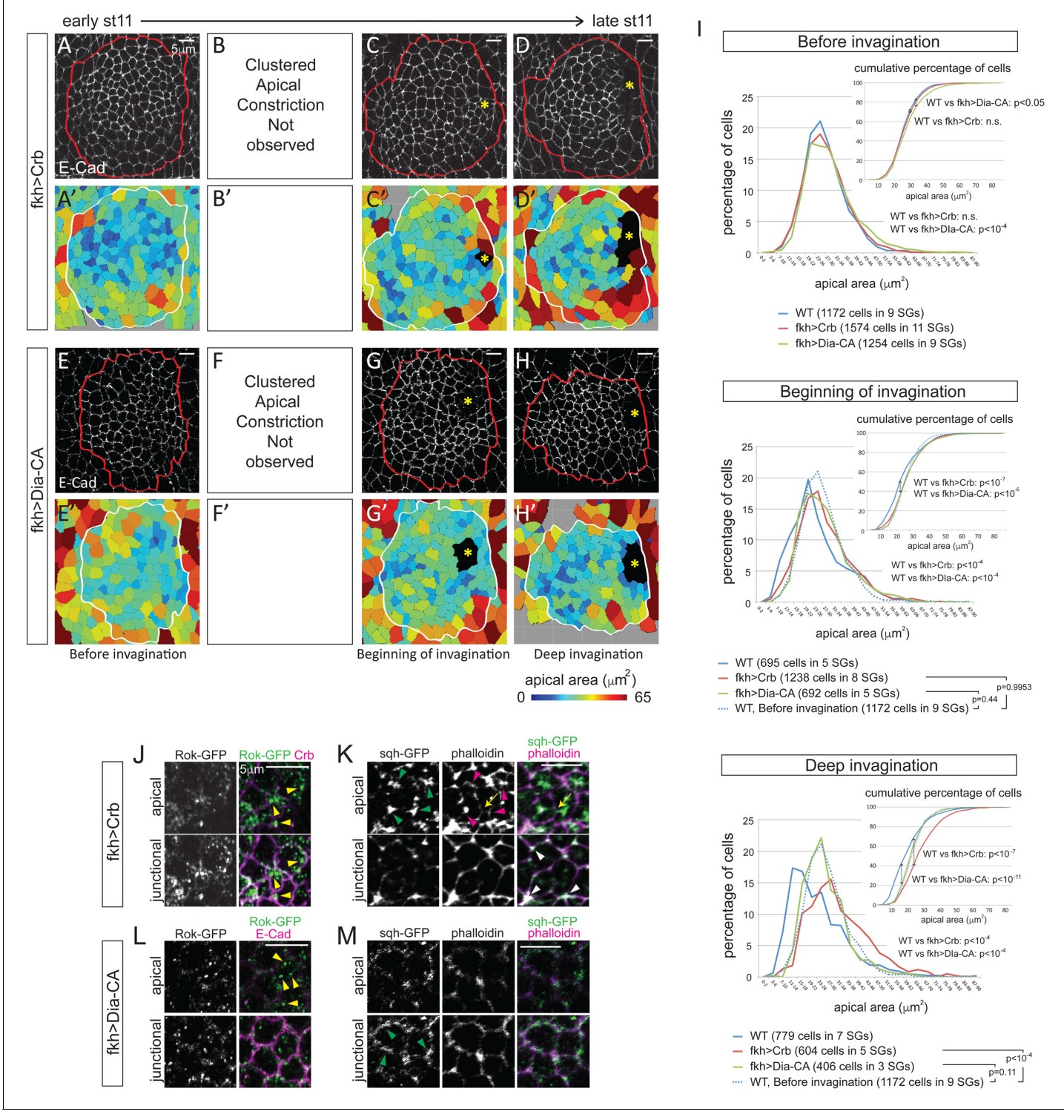

**Figure 5.** Blocking apical constriction does not prevent SG internalization. (A–H) Representative confocal images for Crb- (A–D) and Dia-CA-overexpressing SGs (E–H) at each stage of invagination and the corresponding heat maps for apical area (A'–H'). Red and white lines mark the SGs. Asterisks, invagination pit. (I) Percentage and cumulative percentages of cells with different apical area show a gradual and significant increase of apical area for Crb- and Dia-CA-overexpressing SG cells. P values are calculated using the Mann-Whitney U test (percentage of cells) and using the Kolmogorov-Smirnov test (cumulative percentage of cells). See also *Figure 5—source data 1* and *Figure 5—figure supplement 1*. (J, K) In Crb-overexpressing cells, Rok is observed as small- to medium-sized puncta (yellow arrowheads in J). Strong apicomedial myosin (green arrowheads in J) colocalizes with F-actin (magenta arrowheads in K). Junctional myosin (white arrowheads in K) is also clearly shown. Occasional cell deformation is

*Figure 5 continued on next page*

 Research article                                    Cell Biology | Developmental Biology and Stem Cells

*Figure 5 continued*

observed at the contact site of apicomedial actomyosin complex and the junction (arrows). (L, M) Only small punctate Rok signals are observed in Dia-CA-overexpressing cells (yellow arrowheads in L). Myosin is observed in punctate structures in a broad cortical area (green arrowheads in M).

The following source data and figure supplement are available for figure 5:

**Source data 1.** SG cells quantified for apical area.

**Figure supplement 1.** Overexpression of Crb- and Dia-CA affects the distribution of cells with smaller apical areas.

*H99* outer boundary, indicating loss of compressive forces on cells at and near the SG boundary (*Figure 7F*).

Anisotropic Crb distribution at the placode boundary has been suggested to drive supracellular myosin cable formation during SG formation (*Röper, 2012*). Consistent with this, Crb levels were higher in WT SGs than in surrounding tissues by early stage 11 and further increased by late stage 11, with overt anisotropic localization of Crb in SG cells at the dorsal boundary (*Figure 7H,H',I,I'*). In these cells, myosin levels were very high at the boundary, where Crb levels were lowest (inset). Importantly, Crb levels were lower in *fkh H99* SGs than in WT throughout stage 11 and did not show anisotropy in boundary cells (*Figure 7J,J',K,K'*). Also, unlike in WT SG cells, myosin levels were never high at the dorsal boundary (*Figure 7H'–K'*). Taken together, we conclude that the compressing force generated by the tissue-level supracellular myosin cable, which fails to form in *fkh* mutants where SG levels of Crb are low, contributes to SG invagination.

## Tissue invagination can be decoupled from a tube formation

A range of SG defects was observed in late stage *fog* mutants, with 34% of SGs either partially or fully externalized (*Figure 8A–E, A'–E' and K*). Given that loss of *fog* had relatively minor effects on apical constriction compared to overexpression of Crb or Dia-CA, and that *fog* mutants also form the circumferential cable surrounding the tissue, these severe tube internalization defects were unexpected. Therefore, we asked if SG expression of Fog in otherwise *fog* mutant embryos could rescue internalization using the UAS-Gal4 system, which has allowed the rescue of multiple SG mutant phenotypes, including loss of *fkh* and other transcription factors expressed in the early SG (D. Johnson and D.J. Andrew, unpublished data). The range of defects observed with SG expression of Fog in otherwise *fog* mutant embryos was similar to that seen with complete loss of *fog* (*Figure 7K*), indicating that the Fog requirement for internalization is not SG autonomous. Instead, analysis of early *fog* mutant embryos suggested that the separation of the SG from the surrounding epithelia causes these defects. In stage 11 *fog* mutants, deep ingression furrows often formed between the two tissues, and in more than 60% of embryos with such furrows (24/39), some SG cells at the edge were tilted and even perpendicular to the rest of the cells in the SG epithelium, forming a lip-like structure (*Figure 8F–J and F'–J'*). These cells maintained adhesion to their neighbors and were properly polarized, with their AJs connected to those of neighboring SG cells at the edge. Strong myosin signals at the level of AJs along the edge of the lip (*Figure 8H'*) suggest that pulling or squeezing forces along the edge of the gland in combination with the pushing forces of the ingression furrow caused the SG cells to slip outside the embryo.

We propose that if the ingression furrow formed at or near the posterior end of the primordia (where invagination always initiated), then the resulting glands would slip outside and form fully externalized tubes (*Figure 8J and J'*). However, if the ingression furrows formed elsewhere, the resulting glands would either fully or partially internalize, depending on the extent of ingression between the SG primordia and surface epithelium (*Figure 8I and I'*). Importantly, all *fog* mutant SGs (whether internalized, externalized, elongated, folded or twisted) showed intact, closed and properly polarized tubes with the apical membrane facing an internal lumen (*Figure 8A–E, A'–E'*). The completely externalized SGs that elongated along the embryo surface often formed longer and narrower tubes than internalized SGs (*Figure 8E and E'*), suggesting that, although the SG does not require contact with internal tissues to elongate, both tube diameter and length may be regulated through contact with internal tissues. Altogether, these findings reveal that SG tube formation occurs in a tissue-autonomous manner and can be decoupled from internalization.

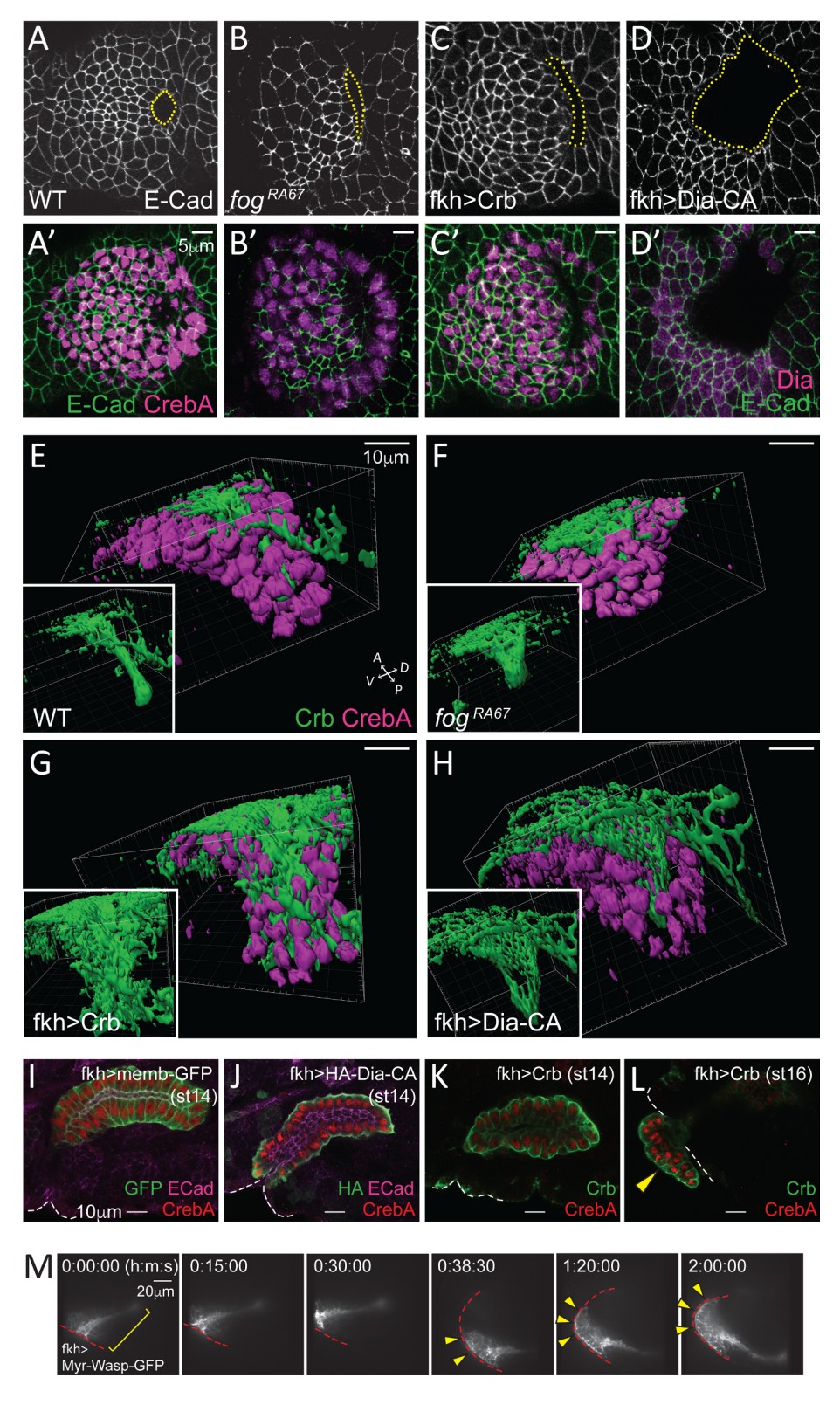

**Figure 6.** Apical constriction is essential for tissue geometry. (A–D, A'–D') Representative images showing morphology and size of the invagination pit (dotted lines) in the different genotypes. (E–H) 3D reconstruction of late stage 11 SGs stained with Crb (green) and CrebA (magenta). Compared to a narrow lumen in a WT SG (E), *fog* mutant (F), Crb-overexpressing (G) and Dia-CA-overexpressing (H) SGs form a wider lumen. Insets, Crb signals only. (I–L) Fully formed late stage SGs. Dia-CA-overexpressing SGs have wider lumens and are closer to the embryo surface (J). Crb-overexpressing SGs

*Figure 6 continued on next page*

*Figure 6 continued*

show normal SG internalization until stage 14 (**K**), but some cells occasionally evaginate at later stages (arrowhead in **L**). White dashed lines, embryo boundary. (**M**) Time-lapse images of Myr-Wasp-overexpressing SG show evagination behavior of SG cells. The SG was re-centered at 38:30. Cells that were completely internalized at the beginning of the movie (bracket) are shown on the embryo surface over time (arrowheads). Red dashed lines, embryo boundary. See *Video 3*.

## Discussion

Apical constriction often occurs as polarized epithelial precursors bend, fold and invaginate to form different tissues and organs (*Martin and Goldstein, 2014*). How tissues/cells undergo apical constriction in a temporally and spatially controlled manner and how much this morphogenetic process contributes to tissue invagination in different developmental contexts have been less clear. Using the *Drosophila* SG as a model allowed us to quantitatively analyze the distribution and magnitude of cell shape changes during invagination and to determine the signaling pathway through which apical constriction is regulated tissue-specifically. Our studies support a model in which apical constriction plays only a minor role in tissues that internalize by budding and that multiple morphogenetic processes contribute to tissue budding.

### Tissue- and cell-specific regulation of apical constriction

Fog has been implicated in invagination of several embryonic tissues, including the ventral furrow, the posterior midgut and the SG, and also functions in imaginal disc folding during larval development (*Manning and Rogers, 2014*; this study). In the ventral furrow, Twist and Snail regulate expression of the Fog ligand and its receptor, respectively (*Manning et al., 2013*; *Seher et al., 2007*). Likewise, Caudal, through activation of additional downstream transcription factors, is required for *fog* expression in the posterior midgut (*Costa et al., 1994*; *Wu and Lengyel, 1998*). Here, we demonstrate that Fkh regulates SG expression of *fog* to coordinate apical constriction of SG cells (*Figure 4*). Therefore, at least during embryogenesis, the tissue-specificity of Fog signaling is determined by the tissue-specific transcription factor(s) that activate expression of *fog* (and its receptors). The downstream pathway components are both maternally deposited and widely expressed, and are thus likely to be shared in all cell/tissue types where Fog functions. Indeed, mutations in RhoGEF2, a key component of the Fog signaling pathway in the mesoderm, cause SG invagination defects (*Nikolaidou and Barrett, 2004*). Due to their ubiquitous expression, localized activation of the downstream components, rather than their localized expression, is likely to determine their site of action. Consistent with this notion, we demonstrated that the robust apicomedial Rok accumulation in the SG cells requires both Fkh and Fog (*Figures 3* and *4*).

An interesting question is what determines regional specificity for Fog signaling within a tissue. Levels of *fog* transcripts are relatively uniform across the primordia, yet both Rok accumulation and apicomedial myosin accumulation occur only in a temporally and spatially regulated subset of SG cells. We propose that production and apical release of Fog and/or the apical localization of its transmembrane receptors could be regulated post-translationally.

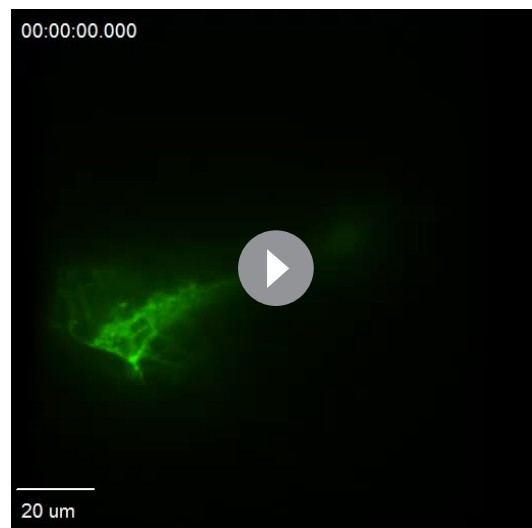

**Video 3.** Myristylated-Wasp-overexpressing SGs evaginate, Related to *Figure 6*. Time-lapse movie for a late stage SG overexpressing a membrane-bound form of Wasp. The sample was re-centered at 38:30. Stacks of two confocal sections are shown. The fully internalized SG at the beginning of the movie gradually evaginates over time. Frames are 5 s apart.

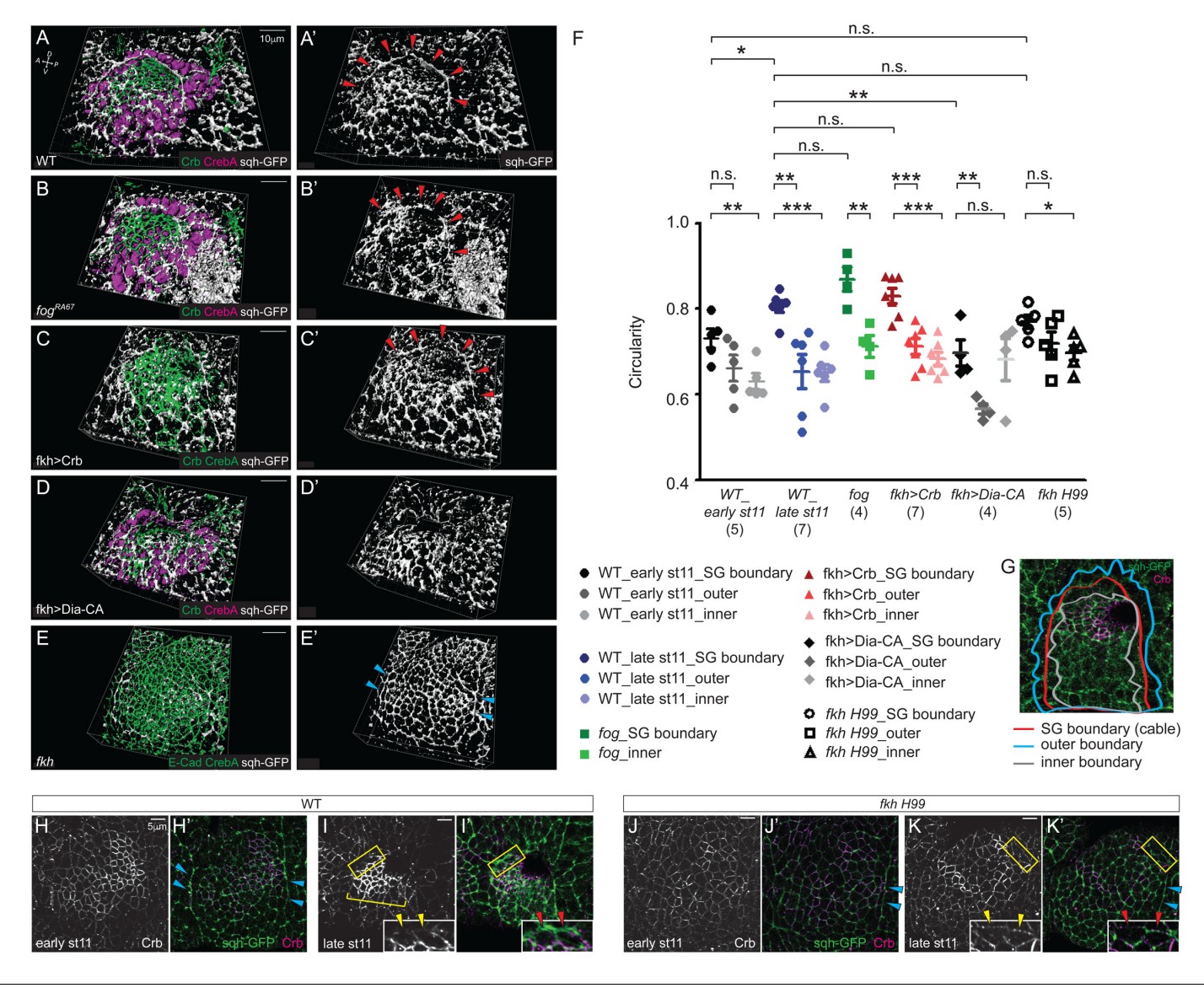

**Figure 7.** The tissue-level supracellular cable is defective in the SGs with defective invagination. (A–E) 3D reconstruction of late stage 11 SGs stained with Crb (green), CrebA (magenta in A, B and D and green in C and E), and sqh-GFP (white). A supracellular myosin cable (red arrowheads) that surrounds the entire SG placode is observed in the WT (A), *fog* mutant (B) and Crb-overexpressing SGs (C). The tissue-level myosin cable is not obvious in the Dia-CA-overexpressing SGs (D). Only the supracellular cables along the lateral boundaries of the placode are shown in the *fkh H99* mutant SGs (cyan arrowheads in E). (A'–E') sqh-GFP signals only. (F, G) Analysis of the circularity of the SG placode boundary as a measure of smoothness and tension. The circularity of the placode boundary in the WT, *fog*, and Crb-overexpressing SGs, where the cable is located, is significantly greater than that of the boundaries shifted outside the placode by one cell (outer boundary) or inside the placode by one cell (inner boundary). Numbers of the SGs measured are shown below each genotype. mean ± SEM; P values are calculated using the unpaired t-test. *p<0.05; **p<0.01; ***p<0.001. (G) An example of a WT SG with the boundaries measured. (H, I) Compared to moderate levels at early stage 11 (H, H'), Crb levels (magenta) notably increase in the SG cells at late stage 11 (bracket). Anisotropic Crb localization (yellow arrowheads) correlates with supracellular myosin cable at the SG boundary (red arrowheads). (J, K) In *fkh H99* mutant SGs, Crb levels in the SG are reduced throughout stage 11. Weak Crb signals do not show anisotropic localization at the SG boundary (yellow arrowheads), and the supracellular myosin cable does not form at the dorsal boundary of the SG (red arrowheads). Cyan arrowheads, supracellular myosin cable along the lateral boundaries.

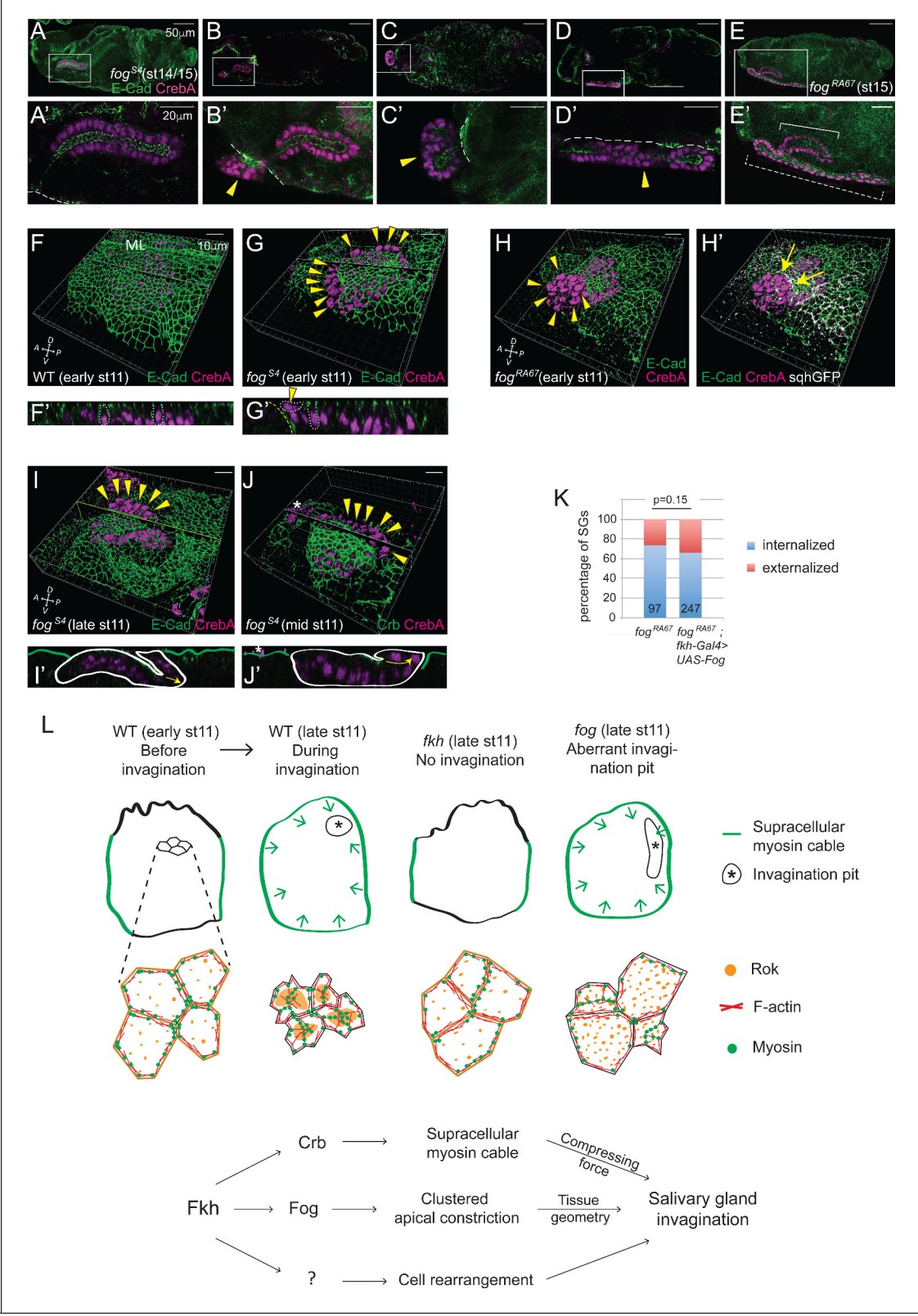

**Figure 8.** Invagination decouples internalization of cells. (A–E, A'-E') *fog* mutant SGs of late stages, from fully internalized (A, A'), partially internalized (B, B') to completely externalized glands, either folded (C, C') or elongated (D, D', E, E'). The externalized SG that migrates along the embryo surface (dashed bracket in E) formed a longer tube than the internalized one (solid bracket in E). All SGs have proper apicobasal polarity. (F, G) Whereas the surface epithelium is in the same plane as the SG in WT (F), a deep ingression between the SG and the neighboring surface epithelium is often

*Figure 8 continued on next page*

*Figure 8 continued*

observed in *fog* mutants (G), with some SG cells perpendicular to the rest of the SG on the apical side of the epithelial sheet (arrowheads). Z sections along the lines (F', G'). Two SG cells are outlined with white dotted lines for each sample; one of them is perpendicularly positioned at the edge in the *fog* mutant (arrowhead). ML, midline. Yellow dashed lines, the boundary of the neighboring surface epithelia. (H) Strong myosin signals are often observed along the edge of the *fog* mutant SG (arrows). Arrowheads, nuclei of the slipped-out cells. (I, J) *fog* mutant SGs with slipped-out cells (arrowheads) in the anterior region of the placode form an externalized tube as cells invaginate (I). *fog* mutant SGs with slipped-out cells in the posterior region of the placode form an externalized tube as cells invaginate (J). Z sections along the lines (I', J'). White lines, the boundary of SGs. Green lines, embryo surface. Arrows, the direction of invagination. Asterisks, non-specific signals. (K) SG-specific expression of Fog in *fog* mutant embryos did not rescue the externalized SG phenotype. Numbers inside bars indicate the number of SGs counted. P value was calculated using the Chi-square test. (L) A model for SG invagination. Fkh-dependent high-level Crb expression in the SG versus low levels of Crb in the surrounding ectodermal tissues regulates formation of the circumferential myosin cable that surrounds the entire SG to provide a compressing force during invagination. Regulation of the SG expression of *fog* by Fkh controls apicomedial localization of Rok and formation of apicomedial myosin, which drives clustered apical constriction to ensure the proper tissue geometry. Fkh-dependent uncharacterized target genes are proposed to drive the cell rearrangement that also contributes to SG internalization.

Excellent candidate regulators include Huckebein (Hkb), an SP1-like transcription factor, and its key target Klarsicht (Klar), a putative regulator of dynein ATPase (*Mosley-Bishop et al., 1999*; *Myat and Andrew, 2000b*, *2002*). The expression of both *hkb* and *klar* is limited to the dorsal posterior domain of the SG, where we observe high Rok, high myosin, and apical constriction. Since Klar is required for the delivery of vesicles to the apical SG surface, which is enriched in minus-end microtubules (MT) (*Myat and Andrew, 2002*), a possible scenario is that during invagination, Klar-dependent apically-targeted vesicles contain Fog ligand and/or its receptor. Consistent with this model, a recent study has demonstrated a requirement for the MT cytoskeleton in the formation of apicomedial myosin (*Booth et al., 2014*). Moreover, unlike WT, *hkb* mutant SGs invaginate from the center of the SG placode (*Myat and Andrew, 2002*).

## Additional morphogenetic processes take place during SG invagination

During ventral furrow formation, where apical constriction plays a major role in tissue invagination (*Guglielmi et al., 2015*), myosin is absent from the cortex and accumulates in only the apicomedial region. Since AJ-associated cortical myosin has been linked to junctional remodeling and cell rearrangement in the intercalating ectodermal cells of the *Drosophila* embryo (*Bertet et al., 2004*; *Blankenship et al., 2006*; *Irvine and Wieschaus, 1994*; *Rauzi et al., 2008*; *Zallen and Wieschaus, 2004*), one might expect that cell rearrangement does not take place during ventral mesoderm formation. Indeed, ventral furrow cells invaginate without changing their position relative to each other; cells constrict predominantly in the ventral/lateral direction, remaining longer along the A/P axis to form a long, narrow furrow along the axis (*Martin et al., 2010*; *Sweeton et al., 1991*).

We show that apical constriction is not required for SG invagination per se, but is needed to acquire proper geometry of the invagination pit and the tube that forms from it (*Figures 5* and *6*). In the SG, where cells sequentially invaginate through a relatively small invagination pit to make a tube, combinatorial forces that drive different morphological changes are expected to contribute to tissue invagination (*Figure 8L*). Correspondingly, in tracheal invagination, at least three independent processes have been implicated: EGF-dependent myosin accumulation (*Brodu and Casanova, 2006*; *Nishimura et al., 2007*), an oriented final cell division (*Kondo and Hayashi, 2013*), and FGF-mediated migratory forces. Only when all three processes are disrupted, do tracheal cells fail to internalize (*Kondo and Hayashi, 2013*). However, since no cell divisions occur once the SG has been specified (*Chung et al., 2014*), oriented cell divisions cannot contribute. Similarly, our finding that the non-autonomous functions of *fog* can lead to fully formed external SGs (*Figure 8*) reduces the likelihood that signaling between the SG and neighboring tissues is critical.

We show at least one additional force contributes to tissue invagination when apical constriction is blocked: the supracellular myosin cable that is under tension and sequentially closes in as SG cells internalize. This circumferential myosin cable forms normally in both *fog* mutant SGs and in Crb-overexpressing SGs, where invagination occurs relatively normally, but fails to form in *fkh* mutant SGs and in Dia-overexpressing SGs, where invagination either fails or is delayed (*Figure 7*). These findings support the idea that tension generated by the myosin cable contributes to and facilitates

invagination (*Figure 7*). This closing process in the SG is reminiscent of the 'purse-strings' observed during wound healing in both flies and vertebrates (*Martin and Parkhurst, 2004*) and also during *Drosophila* dorsal closure (*Franke et al., 2005*). It is notable that multiple forces have been suggested to contribute to dorsal closure; dorsal closure fails only when both the forces of apical constriction of amnioserosa cells and of purse strings from surrounding epidermal cells are ablated (*Kiehart et al., 2000*), although this model of a combinatorial force-component system has been challenged by recent studies showing that the epidermal actin cable tension may not contribute to dorsal closure (*Pasakarnis et al., 2016*).

The difference in invagination defects in Dia-CA-overexpressing SGs (delayed) and *fkh H99* SGs (completely failed), where both apical constriction and the circumferential myosin cable are absent, indicate that additional other forces contribute to SG invagination. We propose that cell rearrangement linked to junctional myosin may also be important. A strong junctional myosin pool is observed throughout SG invagination, with high levels at cell vertices (*Figure 2*; *Röper, 2012*). We have direct and indirect evidence for cell rearrangement in early WT SGs from live imaging and measurements of cell topology (S. Chung, S. Kim and D.J. Andrew, unpublished data), suggesting that this process is linked to tissue invagination. Since myosin localizes in a broader cortical region in Dia-CA-overexpressing SGs (*Figure 5*), it is possible that cell rearrangement of Dia-CA-overexpressing cells is somewhat affected, explaining the delayed invagination and abnormal tissue morphology observed early (*Figure 6D and H*). Nonetheless, cell rearrangement must still occur in the Dia-CA overexpressing glands since they form elongated tubes with an approximately WT arrangement of cells at later stages (*Figure 6J*). The complete failure of invagination in *fkh H99* SGs raises the possibility of failed cell rearrangement. Interestingly, changes in cell topology occur independently of *fkh* function. WT and *fkh H99* mutant SGs show a similar distribution of n-sided cells during early stages of invagination (S. Chung and D.J. Andrew, unpublished data). Consistent with the idea that cortical enrichment of myosin has a role in these topological changes, junctional myosin is intact in *fkh H99* SGs (*Figure 2*). In any case, more direct measurements of cell rearrangement in both WT and *fkh* mutants will be necessary to learn how directed cell rearrangement contributes to SG internalization. Overall, our findings suggest that multiple forces are required for SG invagination and that the system is remarkably robust. Importantly, our finding that polarized and fully elongated tubes can form outside the embryo indicates that SG morphogenesis is largely a tissue-autonomous process.

## Materials and methods

### Fly strains

*fkh⁶ H99* recombinants (RRID:DGGR_130500; RRID:BDSC_1576; *Myat and Andrew, 2000a*) were used for the analysis of *fkh* mutant SGs. UAS-Crb (RRID:DGGR_108289; *Wodarz et al., 1995*), UAS-HA-Dia-CA (RRID:BDSC_27616; *Somogyi and Rørth, 2004*) and UAS-Myr-Wasp-GFP (C-terminal GFP tag added to construct described in FBal0243602) driven by *fkh-Gal4* (FBtp0013253/ FBti0027904; *Henderson and Andrew, 2000*) were used. *fkh-Gal4* driving *UAS-Tmem-GFP* (Ftp0011013 [aka *UAS-GFP-CAAX*]; *Kakihara et al., 2008*) was used as a control. Two null alleles for *fog* (*fog^{S4}* and *fog^{RA67}*; RRID:DGGR_106665; RRID:BDSC_6218) were analyzed; both resulted in indistinguishable phenotypes. *UAS-fog* (FBtp0021363; *Dawes-Hoang et al., 2005*) driven by *fkh-Gal4* in *fog^{RA67}* embryos at 25°C was used for SG-specific rescue experiment of *fog*. *sqh^{AX3}*; *sqh-GFP* (RRID:BDSC_57144; *Royou et al., 2004*) and *sqh-Rok-GFP* (RRID:BDSC_52289; *Abreu-Blanco et al., 2014*) were used to analyze myosin and Rok accumulation, respectively.

### Antibody staining and fluorescent in situ hybridization

For most samples, dechorionated embryos were fixed in 1:1 heptane:formaldehyde for 40 min and devitellinized with 80% EtOH. For phalloidin staining, embryos were hand-devitellinized. Antibodies used include: mouse α-Crb (RRID:AB_528181; DSHB; 1:10), rabbit α-CrebA (RRID:AB_10805295; *Fox et al., 2010*; 1:3000), chicken α-GFP (RRID:AB_300798; Abcam; 1:1000), rat α-E-Cad (AB_528120; DSHB; 1:10), rabbit α-Dia (*Afshar et al., 2000*; 1: 5000), mouse α-HA (RRID:AB_514506; Roche; 1:50), guinea pig α-Sage (RRID:AB_2632603; *Fox et al., 2013*); 1:100), rabbit α-Fkh (RRID:AB_2632955; *Fox et al., 2013*; 1:2000), rabbit α-β-Gal (AB_788167; Novus; 1:1000). Alexa fluor 488, 555, 647-labelled secondary antibodies were used at 1:200 (Molecular probes: RRID:AB_

142924; RRID:AB_141822; RRID:AB_142754; RRID:AB_141693; RRID:AB_141373; RRID:AB_141778; RRID:AB_143165; RRID:AB_162543). Alexa fluor-546-labeled phalloidin was used for F-actin labelling (Molecular Probes: RRID_AB2632953; 1:250). Fluorescent in situ hybridization with an RNA probe specific for *fog* and antibody staining with anti-Sage was performed as described (*Knirr et al., 1999*). For the in situ analysis, a PCR-labelled product of the *fog* ORF was made from cDNA generated by reverse transcription of total RNA using the following primers: fog-5': ATGTCTCCGCCCAA TTGTCT and fog-3': GATGACTGAAAAGCGGCGGC. All images were taken with a Zeiss LSM 780 confocal microscope.

## Scanning electron microscopy

Dechorinated embryos were fixed in 25% glutaraldehyde in cacodylate buffer and hand-devitellinized. Embryos were post-fixed in 1% $OsO_4$ and dehydrated through an EtOH series. They were dried using Hexamethyldislazane, coated with gold palladium, and examined and photographed in a LEO/Zeiss Field-emission SEM.

## Cell segmentation and apical area calculation

E-Cad/CrebA stained SGs were imaged with a Zeiss LSM 780 confocal microscope. Two or three focal planes (0.37 µm apart) of apical domain were used to generate a maximum intensity projection (Zeiss Zen software; Germany). Cell segmentation was performed along the E-Cad signals and the apical area for each cell was calculated using the Bitplane Imaris program. During 'Before invagination' and 'Clustered apical constriction', all cells were at the surface in the same focal plane. During 'Beginning of invagination', a few cells in the posterior/dorsal region of the placode were found in a basal position relative to neighboring cells (at 0.37–1.11 µm deeper focal planes). 'Deep invagination' included samples with cells that had internalized >2 µm.

## Frequency distribution and scatter plots for SG cells with different apical area

Frequency distribution of cells with a different apical area (bin width = 4) was performed using a GraphPad Prism program. For median scaling normalization of *fkh H99* SG cells (*Figure 1K*), the apical area of each cell was multiplied by 1.2.

The x and y coordinates for each cell, which correspond to the position along the A/P and D/V axis of the tissue, respectively, were obtained using Imaris. The position of SG cells relative to the center of the invagination pit was calculated and plotted. Standard deviations for the positions along the A/P or D/V axis were compared to show dispersion of cells.

## Quantification of myosin intensity

Cells in the posterior/dorsal region of the placode were chosen for quantification. For all quantifications, maximum intensity projections that span the apical and the junctional region of SG cells were used (Zeiss Zen software) and measurements were performed with Fiji software. Regions were drawn manually along the inner or outer boundary of E-Cad signals of each cell to calculate the apical and junctional intensity. Integrated density for myosin signals was measured after background correction and mean values from three individual measurements were used.

## Negative correlation between apical area and myosin intensity

Cell segmentation for six WT SGs at deep invagination was performed as described above. Intensity mean was measured for the entire myosin signals for each segmented cell and re-scaled for 0 to 100 (a.u.) for plotting. Correlation (Pearson) and P values were calculated using the GraphPad Prism software.

## Live imaging

*sqh^AX3*; *sqh-GFP* or *sqh-GFP-Rok* embryos were dechorinated and mounted ventral side up on a glass slide coated with heptane glue (double-sided tape soaked in heptane). Spacer coverslips (No. 1.5) were attached to prevent embryos from being squeezed. Halocarbon oil mixture 27:700 (1:1) was added on top of the embryos. A No. 1 coverslip was placed on top and sealed with nail polish. Images were taken every 5s with a 3i Marianas/Yokogawa Spinning Disk Confocal microscope to

capture apicomedial myosin pulsing and every 20 s with a Zeiss LSM 780 confocal microscope to capture apicomedial Rok signals.

## Rok inhibitor treatment

$sqh^{AX3}$; sqh-GFP embryos were collected and aged until stage 9. Dechorinated embryos were immersed for five minutes in a 1:10 dilution of Citrasolv in water (Citra-Solv, Danbury, CT) to render embryos permeable (*Rand et al., 2012*). Embryos were rinsed with water and PBS several times and were incubated in Y-27632 (500 µM in PBS) for three hours. Embryos were fixed as described above and immuno-stained.

## 3D reconstruction of SGs

3D reconstruction of SGs was performed with the confocal stacks using the Imaris program.

## Quantification of circularity and smoothness

Cell boundaries were manually traced in ImageJ to quantify the length and area of the boundary corresponding either to the SG placode boundary or the boundary shifted one cell layer outside or inside the placode boundary. The ventral midline of the embryo was used as the ventral boundary of the traced circle in all cases. For *fog* mutants, only circularity of the SG boundary and inner boundary was measured, since the cells in the neighboring epithelia were often out of the focal plane due to the twisted phenotype of the embryos. Circularity was calculated based on the fact that for a perfect circle the circularity $C = 1 = 4 \pi$ area/perimeter$^2$ (*Lawrence et al., 1999*). Circularity values were plotted in Prism and statistical significance in differences in the circularity of outer and inner boundaries compared to the circularity of the cable was analyzed using a two-tailed t test.

## Acknowledgements

We thank Bloomington, E Chen, A Martin and S Parkhurst for fly stocks, Flybase for information on gene structure, and C Hanlon, R Loganathan, C O'Kane and P Paul for their helpful comments on the manuscript.

## Additional information

### Funding

| Funder | Grant reference number | Author |
|---|---|---|
| National Institutes of Health | RO1 DE13899 | Deborah J Andrew |

The funders had no role in study design, data collection and interpretation, or the decision to submit the work for publication.

### Author contributions

SYC, Conceptualization, Data curation, Formal analysis, Investigation, Visualization, Methodology, Writing—original draft, Writing—review and editing; SK, Investigation, Methodology; DJA, Conceptualization, Formal analysis, Supervision, Funding acquisition, Validation, Writing—original draft, Project administration, Writing—review and editing

### Author ORCIDs

SeYeon Chung, http://orcid.org/0000-0002-5493-6424
Deborah J Andrew, http://orcid.org/0000-0003-1051-6935

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
