## [Decision Letter]

[Editors’ note: this article was originally rejected after discussions between the reviewers, but the authors were invited to resubmit after an appeal against the decision.]

Thank you for choosing to send your work, "Uncoupling apical constriction from tissue invagination", for consideration at *eLife*. Your initial submission has been assessed by a Senior Editor in consultation with a member of the Board of Reviewing Editors (Hugo Bellen), and two reviewers. Although the work is of interest, we regret to inform you that the findings at this stage are too preliminary for further consideration at *eLife*.

As you will notice one reviewer is very positive, while one reviewer is positive, but feels that additional data are required to warrant publication in *eLife*. I am ambivalent. Can you let me know if you can address the comment: "Given the extensive descriptions in the Roper paper, however, and the limited treatment of the cable in the present paper, a more careful mechanical treatment, including quantification of the cable and cell shapes, couple with biophysical modeling would be required to justify publication in *eLife*". Please let me know if you can and are willing to address this issue, and if so, how. I do not feel that modeling is crucial but the first portion of the comment seems important to me.

*Reviewer #1:*

The critical role of apical constriction in cell rearrangements from gastrulation onward has been recognized for decades but the mechanisms underlying it have remained more elusive. The last six years have seen a revolution in our thinking about the mechanisms by which apical constriction is driven, based in large part on studies of the *Drosophila* invaginating mesoderm. This has revealed a novel and unexpected role for an apical myosin network created in response to cell-cell signaling and regulated by apical Rok. However, the generality of this model, in particular in invaginating tubular organs, has remained unexamined. Here Chung et al. present a stunning examination of this, using the extremely tractable *Drosophila* salivary gland as a model. In a series of experiments combining sophisticated genetic tools, beautiful imaging and extensive quantitative analysis, they lay out a pathway leading from the transcription factor Fkh through the cell signaling molecule Fog to Rok and myosin. They also find some exceptionally surprising things along the way-most particularly that the system is remarkably robust, such that eliminating apical construction does not block organ invagination. I think these results will be of very broad interested to both cell and developmental biologists.

I had some questions and suggestions that I think would strengthen the story.

Figure 1. I found the dual 3D images here very informative, as they give insight into how apical constriction combines with other cell rearrangements. Could the authors add one of an intervening stage as the pit begins to form? Also, would it be possible to create similar images of the *fog*, crb- and Dia-CA expressing embryos?

Figure 1. I found it intriguing that there seem to be two clusters of invaginating cells, anterior and posterior, only the posterior of which goes into the pit. This might be worth noting.

Subsection “Blocking apical constriction does not arrest SG invagination”, last paragraph. The authors state that "the significant 'wiggliness' of the AJs suggests a decrease of cortical tension." I think this is unlikely-in cultured mammalian cells, we and others have found that decreasing cortical tension (e.g., by Rok or myosin inhibition) leads to smoothly rounded cell borders – really wiggly cell borders are likely to reflect actin filaments oriented perpendicular to the membrane.

Subsection “Blocking apical constriction does not arrest SG invagination”. When during the process of convergent elongation of the organ primordium does Fog expression go off?

Subsection “Blocking apical constriction does not arrest SG invagination”. In interpreting their Dia-CA experiments the authors might take a look at Homem and Peifer (Development. 2008 Mar;135(6):1005-18) in which it was observed that Dia-CA increased cortical myosin in the amnioserosa and altered apical constriction.

The only set of data I found was weak was the attempt to rescue Fog by SG specific Fog re-expression. Given the likelihood that levels and timing were not precisely replicated, I think the speculation about non-autonomous functions was not well supported and I would suggest removing this data.

Discussion. The authors mention a number of times the incredibly provocative fact that they may have identified a GPCR related to Mist expressed in the SG. This would be very interesting, but the degree of discussion of this unpublished data in the manuscript seemed unwarranted unless they wanted to add this data.

Subsection “Additional morphogenetic processes take place during SG invagination”, last paragraph. The authors should add the data showing that the tissue level circumferential myosin cable does form in *fog* mutants – this is a key part of their model. More generally, I think a more careful description of the progression of junctional myosin in wildtype and mutants would be beneficial – it appeared, for example, that it was AP planar polarized before the process began.

The final circuitry diagram in Figure 7 is nice but would be better if supplemented with a diagram showing the cell shape changes themselves and how the authors think apical constriction and tissue level constriction work together.

*Reviewer #2:*

The manuscript describes the cell biology behind the invagination of salivary gland primordia in the *Drosophila* embryo. The changes in cell morphology that occur during this process has been published previously, but the present paper incorporates new features (e. g., the pulsatile apical medial myosin networks) that allow comparison to other well characterized morphogenetic movements in *Drosophila*, especially the ventral furrow and posterior midgut. In many respects (e. g., the involvement of RoK and Folded gastrulation) the SC pathway shows a regulation of apical constriction and myosin activity very similar to these other processes.

Mutations in Fork head disrupt the invagination process, although they allow apparent initial specification of the primordia, given the CrebA marker expression. Some of this has been previously shown, but the authors extend their earlier phenotypic characterizations by blocking the associated apoptosis using the H99 deficiency. Under these conditions, the pulsing medial myosin population observed in the SG of wild type embryos is not observed, nor does Rok-GFP accumulate in the large globular structures observed in the posterior SG of wildtype There is also no Fog expression in mutants, and pattern of apical constriction is randomized. Conclusions based on these phenotypes are difficult, however, given the uncertainty about how normal or unnormal cells blocked in apoptosis are, and where exactly Fork head intervenes in the hierarchy between programming and cell behavior. Fork head's role is more extensive than just apical constriction, given that the mutants apparently fail to form complete actin cables surrounding the primordial (poorly documented, but see below).

To inhibit all apical constriction, the authors use Crb- or Dia-CA-overexpression. These manipulations are associated with increased apical area (i.e., decreased apical constriction), but surprisingly, even in the most extreme cases, do not block SG invagination. The invaginations are abnormally shaped, however, elongated along the DV axis. The authors attribute these defects to the failure of apical constriction, but overexpression of Crb and Dia may have complex cumulative effects not related to their earlier effects on apical constriction. The most interesting result here, however, as emphasized in the paper's title, is that even in these extreme circumstances, invagination still occurs. The paper would be much stronger if the authors had developed experiments that actually address in a quantitative rigorous way the underlying mechanism that accomplished this invagination.

One potential mechanism has in fact been proposed earlier by Roper (Developmental Cell 2012) and involves a supracellular myosin cable surrounding the SG primordium. Contraction of this cable could act as a "purse string" that would drive internalization of the primordium. By blocking apical constriction, the present paper provides additional arguments for the importance of this cable. Given the extensive descriptions in the Roper paper, however, and the limited treatment of the cable in the present paper, a more careful mechanical treatment, including quantification of the cable and cell shapes, couple with biophysical modeling would be required to justify publication in *eLife*. In the Discussion, the authors mention another mechanism that might also contribute to invagination (myosin accumulation at cell vertices) but these possibilities need to be developed in greater experimental and theoretical detail.

Overall, I think this paper is a very solid contribution to understanding salivary gland morphogenesis, but does not contain sufficiently new unexpected experimental observations or general concepts. This cannot be improved by simple adding a few additional experiments and in its present form is more suitable to a more specialized journal.

---

## [Author Response]

[Editors’ note: the author responses to the first round of peer review follow.]

*Reviewer #1:*

*[…] I had some questions and suggestions that I think would strengthen the story.*

*Figure 1. I found the dual 3D images here very informative, as they give insight into how apical constriction combines with other cell rearrangements. Could the authors add one of an intervening stage as the pit begins to form? Also, would it be possible to create similar images of the fog, crb- and Dia-CA expressing embryos?*

This was an excellent suggestion. We have added both an intervening WT stage to Figure 1 and similar images of the other genotypes to Figure 6.

*Figure 1. I found it intriguing that there seem to be two clusters of invaginating cells, anterior and posterior, only the posterior of which goes into the pit. This might be worth noting.*

The anterior domain of apical constriction corresponds to a slight depression in the gland that forms in the anterior domain after the posterior cells have internalized – visible during stage 12. We have added a comment in the Results on the anterior domain of apical constriction.

*Subsection “Blocking apical constriction does not arrest SG invagination”, last paragraph. The authors state that "the significant 'wiggliness' of the AJs suggests a decrease of cortical tension." I think this is unlikely-in cultured mammalian cells, we and others have found that decreasing cortical tension (e.g., by Rok or myosin inhibition) leads to smoothly rounded cell borders – really wiggly cell borders are likely to reflect actin filaments oriented perpendicular to the membrane.*

The paper by Choi et al. (2016), which we now also cite, indicates that increased wiggliness of the AJs in cultured mammalian cells (observed at the level of confocal microscopy of AJ markers as done here) corresponds to a decrease in cortical tension with multiple genetic manipulations, consistent with findings from other groups.

*Subsection “Blocking apical constriction does not arrest SG invagination”. When during the process of convergent elongation of the organ primordium does Fog expression go off?*

We can detect fog transcripts through early stage 14, when the tube is still elongating. We have added this information to the paper.

*Subsection “Blocking apical constriction does not arrest SG invagination”. In interpreting their Dia-CA experiments the authors might take a look at Homem and Peifer (Development. 2008 Mar;135(6):1005-18) in which it was observed that Dia-CA increased cortical myosin in the amnioserosa and altered apical constriction.*

We have now included a citation of this paper since similar increases in cortical myosin are observed in both systems, although the consequences are not the same in the two tissues.

*The only set of data I found was weak was the attempt to rescue Fog by SG specific Fog re-expression. Given the likelihood that levels and timing were not precisely replicated, I think the speculation about non-autonomous functions was not well supported and I would suggest removing this data.*

We would like to include this data, since we have been able to rescue every other SG mutation we have ever tested using the same system, including rescuing *fkh* mutant SGs (unpublished work from a student in the lab). We have added this information to the paper.

*Discussion. The authors mention a number of times the incredibly provocative fact that they may have identified a GPCR related to Mist expressed in the SG. This would be very interesting, but the degree of discussion of this unpublished data in the manuscript seemed unwarranted unless they wanted to add this data.*

We have removed mention of the Mist-related GPCR.

*Subsection “Additional morphogenetic processes take place during SG invagination”, last paragraph. The authors should add the data showing that the tissue level circumferential myosin cable does form in fog mutants – this is a key part of their model. More generally, I think a more careful description of the progression of junctional myosin in wildtype and mutants would be beneficial – it appeared, for example, that it was AP planar polarized before the process began.*

We have now analyzed the tissue level circumferential myosin cable in all of the genotypes used in this study – new Figure 7. We have also demonstrated loss of high Crb levels and loss of its anisotropic distribution in SG border cells in fkh H99 mutants. These findings are consistent with those of Röper (2012) linking anisotropic Crb distribution to cable formation.

*The final circuitry diagram in Figure 7 is nice but would be better if supplemented with a diagram showing the cell shape changes themselves and how the authors think apical constriction and tissue level constriction work together.*

We have now included such a circuitry diagram in the final panel of the last figure.

*Reviewer #2:*

*The manuscript describes the cell biology behind the invagination of salivary gland primordia in the Drosophila embryo. The changes in cell morphology that occur during this process has been published previously, but the present paper incorporates new features (e. g., the pulsatile apical medial myosin networks) that allow comparison to other well characterized morphogenetic movements in Drosophila, especially the ventral furrow and posterior midgut. In many respects (e. g., the involvement of RoK and Folded gastrulation) the SC pathway shows a regulation of apical constriction and myosin activity very similar to these other processes.*

*Mutations in Fork head disrupt the invagination process, although they allow apparent initial specification of the primordia, given the CrebA marker expression. Some of this has been previously shown, but the authors extend their earlier phenotypic characterizations by blocking the associated apoptosis using the H99 deficiency. Under these conditions, the pulsing medial myosin population observed in the SG of wild type embryos is not observed, nor does Rok-GFP accumulate in the large globular structures observed in the posterior SG of wildtype There is also no Fog expression in mutants, and pattern of apical constriction is randomized. Conclusions based on these phenotypes are difficult, however, given the uncertainty about how normal or unnormal cells blocked in apoptosis are, and where exactly Fork head intervenes in the hierarchy between programming and cell behavior. Fork head's role is more extensive than just apical constriction, given that the mutants apparently fail to form complete actin cables surrounding the primordial (poorly documented, but see below).*

H99 deficiency SGs develop completely normally. This is now cited in the Results (Myat and Andrew, 2000a).

We agree that Fkh plays a major role in multiple aspects of SG development and physiology, but, importantly, SGs are specified in *fkh* mutants, based on observing normal SG expression of approximately 40% of known SG genes in *fkh* nulls (from in situ analysis of more than 100 known SG-expressed transcripts, Maruyama et al., 2011).

The failure to form myosin cables around the SG primordia in *fkh* mutants is now documented (along with an analysis of the myosin cables of all the genotypes analyzed in this paper).

*To inhibit all apical constriction, the authors use Crb- or Dia-CA-overexpression. These manipulations are associated with increased apical area (i.e., decreased apical constriction), but surprisingly, even in the most extreme cases, do not block SG invagination. The invaginations are abnormally shaped, however, elongated along the DV axis. The authors attribute these defects to the failure of apical constriction, but overexpression of Crb and Dia may have complex cumulative effects not related to their earlier effects on apical constriction. The most interesting result here, however, as emphasized in the paper's title, is that even in these extreme circumstances, invagination still occurs. The paper would be much stronger if the authors had developed experiments that actually address in a quantitative rigorous way the underlying mechanism that accomplished this invagination.*

*One potential mechanism has in fact been proposed earlier by Roper (Developmental Cell 2012) and involves a supracellular myosin cable surrounding the SG primordium. Contraction of this cable could act as a "purse string" that would drive internalization of the primordium. By blocking apical constriction, the present paper provides additional arguments for the importance of this cable. Given the extensive descriptions in the Roper paper, however, and the limited treatment of the cable in the present paper, a more careful mechanical treatment, including quantification of the cable and cell shapes, couple with biophysical modeling would be required to justify publication in eLife. In the Discussion, the authors mention another mechanism that might also contribute to invagination (myosin accumulation at cell vertices) but these possibilities need to be developed in greater experimental and theoretical detail.*

We agree that the inclusion of a full analysis of the myosin cable and associated cell shapes is an important addition to this study. We have done such an analysis for all of the genotypes included in this study. We have also demonstrated loss of SG expressed anisotropic Crb in *fkh* H99 SGs, providing support for and extending the molecular pathway proposed by Röper, 2012.